# Fire-derived organic matter retains ammonia through covalent bond formation

Rachel Hestrin[1], Dorisel Torres-Rojas[1], James J. Dynes [2], James M. Hook[3], Tom Z. Regier[2], Adam W. Gillespie[2,6], Ronald J. Smernik[4] & Johannes Lehmann [1,5]

Fire-derived organic matter, often referred to as pyrogenic organic matter (PyOM), is present in the Earth's soil, sediment, atmosphere, and water. We investigated interactions of PyOM with ammonia ($NH_3$) gas, which makes up much of the Earth's reactive nitrogen (N) pool. Here we show that PyOM's $NH_3$ retention capacity under ambient conditions can exceed 180 mg N g$^{-1}$ PyOM–carbon, resulting in a material with a higher N content than any unprocessed plant material and most animal manures. As PyOM is weathered, $NH_3$ retention increases sixfold, with more than half of the N retained through chemisorption rather than physisorption. Near-edge X-ray absorption fine structure and nuclear magnetic resonance spectroscopy reveal that a variety of covalent bonds form between $NH_3$-N and PyOM, more than 10% of which contained heterocyclic structures. We estimate that through these mechanisms soil PyOM stocks could retain more than 600-fold annual $NH_3$ emissions from agriculture, exerting an important control on global N cycling.

[1] Soil and Crop Sciences, School of Integrative Plant Science, Bradfield Hall, Cornell University, Ithaca, NY 14853, USA. [2] Canadian Light Source Inc., 44 Innovation Boulevard, Saskatoon, SK S7N 2V3, Canada. [3] NMR Facility & Spectroscopy Lab, Mark Wainwright Analytical Centre and School of Chemistry, University of New South Wales, Sydney, NSW 2052, Australia. [4] School of Agriculture, Food and Wine, The University of Adelaide, Waite Campus, Urrbrae, SA 5064, Australia. [5] Atkinson Center for a Sustainable Future, Rice Hall, Cornell University, Ithaca, NY 14853, USA. [6] Present address: School of Environmental Sciences, University of Guelph, Guelph N1G 2W1 ON, Canada. Correspondence and requests for materials should be addressed to J.L. (email: CL273@cornell.edu)

The Earth's soil, atmosphere, marine sediment, and ocean water contain large quantities of pyrogenic C (54–109, 0.26 $10^{-3}$, 480–1440, and 26–145 Pg of C, respectively[1,2]). In soil, most of this pyrogenic C originates from burnt biomass generated during vegetation fires, which contributes up to 129 Tg yr$^{-1}$ of PyOM–carbon (PyOM–C) to soil C stocks[1]. Many aspects of pyrogenic C biogeochemistry remain poorly understood, including interactions between pyrogenic C's heterogeneous surface—containing both aromatic and aliphatic C, condensates, and other elements such as N, H, and O—and environmental N sources. Interactions between PyOM and environmental N may influence gaseous N emissions, N leaching, N availability to living organisms, and global N transport[3]. Here, we focus on PyOM's interactions with $NH_3$—the atmosphere's most abundant alkaline gas. Global $NH_3$ emissions are projected to double by 2050 and constitute a large part of the Earth's reactive N pool[4–6]. Common sources of $NH_3$ in soils include decomposing organic matter, rainwater, and N fertilizer. Laboratory studies show that various forms of natural and industrially modified organic matter can retain $NH_3$[7–18], but the $NH_3$ retention capacity of natural PyOM stocks and the mechanisms responsible for $NH_3$ retention under ambient conditions have not been established. Therefore, the extent to which these studies can inform our understanding of PyOM's role in global biogeochemical cycles is unknown.

Proposed mechanisms for $NH_3$ retention by natural PyOM include physisorption, electrostatic interactions, and precipitation of ammonium ($NH_4^+$) salts[7–9]. Although these retention mechanisms would allow PyOM to act as a temporary N sink, N retained in these ways would be readily available for plant and microbial uptake, or loss through gas or solute transport[7]. Conversely, the formation of stronger covalent bonds between PyOM and $NH_3$ would result in more persistent N retention, allowing PyOM to serve as a dynamic, long-term N source and sink—both capturing $NH_3$ from its surroundings and slowly releasing it over time. This could also result in greater coupling of global C and N cycling, as covalently bound $NH_3$–N would be carried with the PyOM–C as it traveled over great distances[1]. However, until now, covalent bond formation between natural PyOM and $NH_3$ under ambient terrestrial conditions has not been observed.

Some evidence exists that under certain laboratory conditions, covalent bonds can form between $NH_3$ and industrially produced relatives of PyOM or secondary organic aerosols found in the atmosphere. Graphene oxides and activated carbons can form a variety of cyclic and non-cyclic N structures when exposed to $NH_3$ at temperatures exceeding 200 °C[10–14,17,18]. However, these materials are often modified (e.g., through exposure to chemical oxidants and heat or impregnated with metals), differ considerably from natural PyOM in surface area and functional group composition, and are exposed to $NH_3$ under conditions that are not representative of the natural environment[19–21]. Thus, it is unknown whether these studies of graphene oxides and activated carbons can be used to predict interactions between natural PyOM and $NH_3$, and whether the same variety of covalent N structures would develop under natural environmental conditions. Following exposure to $NH_3$ at ambient temperatures, industrial relatives of PyOM can form non-cyclic amine and amide bonds[22]. Secondary organic aerosols found in the atmosphere—which contain functional groups present in terrestrial PyOM—can form both non-cyclic N structures as well as N heterocycles following exposure to $NH_3$, $NH_4^+$, and amino acids under atmospheric conditions[23,24]. However, the formation of aromatic and non-aromatic heterocyclic N structures between terrestrial PyOM and $NH_3$ has never been observed under ambient environmental conditions. This is of great interest because enrichment with these heterocyclic N structures

influences the electrochemical properties, absorptive capacity, and environmental persistence of both natural and industrial pyrogenic C materials[16,22–27]. If heterocyclic N structures develop between PyOM and $NH_3$ under natural conditions, this interaction would have important consequences for global C and N cycling.

In order to assess the impact of PyOM stocks on global nutrient cycles, it is also necessary to consider PyOM's dynamic nature. Similar to the variety found in other sources of organic matter, different types of PyOM have different physical and chemical characteristics, including elemental makeup, functional group composition, surface area, pH, and other properties[28]. Additionally, PyOM properties change over time, as the material is exposed to water, sunlight, microbial activity, and other oxidizing forces[29–31]. Such variation in physiochemical properties can drastically alter PyOM's role in the environment. Thus, to understand the influence of PyOM–$NH_3$ interactions on global N cycling, it is important to consider how PyOM's $NH_3$ retention capacity might change over time. In this study, we investigate PyOM's $NH_3$ retention capacity under ambient conditions, N retention mechanisms, and whether retention capacity develops as PyOM stocks are weathered. We find that PyOM retains a surprising quantity of $NH_3$–N and that this retention capacity increases significantly as PyOM is exposed to conditions mimicking natural weathering processes. More than half of the $NH_3$–N is retained through chemisorption, including the formation of a variety of covalent bonds. We estimate that through these mechanisms soil PyOM stocks could play an important role in the global N cycle.

## Results

**Weathering increases PyOM N retention capacity**. PyOM produced from woody biomass was oxidized to generate a gradient of weathered PyOM[30,31] and subsequently exposed to $NH_3$ vapor at ambient temperature and pressure (35 °C and 80–800 Torr). Total $NH_3$ capture increased more than sixfold after oxidation, from 2.3 mmol g$^{-1}$ PyOM–C in unoxidized PyOM to 13.5 mmol g$^{-1}$ PyOM–C in highly oxidized PyOM (Fig. 1a), showing that PyOM can retain substantial quantities of N from this form of $NH_3$. Although specific surface area (SSA) and low pH may contribute to the $NH_3$ retention capacity of some pyrogenic C materials[7,8,32], these characteristics did not explain the trends observed here. PyOM SSA decreased with oxidation and therefore could not have contributed to the increase in $NH_3$ retention observed in highly oxidized PyOM samples (Fig. 1b). PyOM pH also decreased with oxidation (Fig. 1c). However, when unoxidized PyOM was incubated with hydrochloric acid, which lowered its pH without altering key oxygen-containing functional groups (Supplementary Fig. 1), $NH_3$ retention remained unchanged (Supplementary Fig. 2). This shows that although oxidation and low pH are correlated, pH itself did not drive $NH_3$ retention and cannot be used to predict PyOM oxidation or $NH_3$ retention capacity. Instead, our analyses indicate that functional group composition may be a more reliable determinant of PyOM's $NH_3$ retention capacity[11,21]. Peak height ratios measured by Fourier transform infrared spectroscopy (FTIR) and integrated peak areas measured by solid-state $^{13}C$ nuclear magnetic resonance (NMR) spectroscopy suggest that with progressive oxidation, PyOM's oxygen-containing functional groups increase relative to aromatic C structures (Fig. 1d and Supplementary Figs. 3 and 4). Quantitative stoichiometric measurements show that an increase in PyOM O:C ratio corresponds with the same trends observed through these spectral analyses. Taken together, these results highlight PyOM's substantial and dynamic N retention capacity, and the relevance of weathering

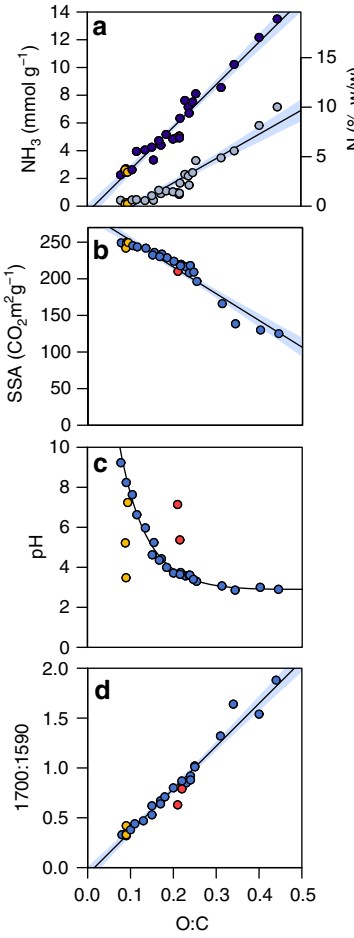

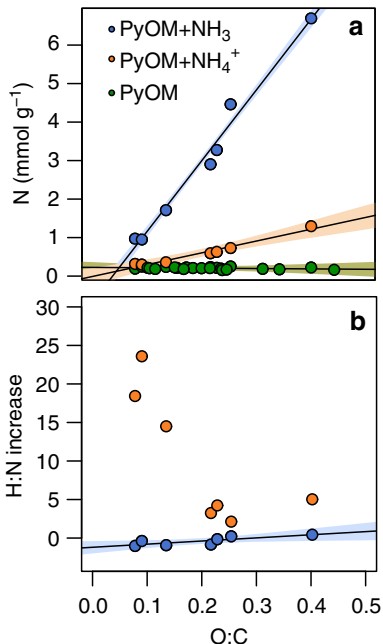

**Fig. 2** Pyrogenic organic matter N content and H:N ratio following exposure to ammonium or ammonia. **a** Pyrogenic organic matter (PyOM) nitrogen (N) retention in mmol g$^{-1}$ PyOM–carbon (C) measured by dry combustion is significantly associated with molar O:C ratios following exposure to NH$_3$ and NH$_4^+$ ($y = 18.25x - 0.0665$, $p < 0.001$ and $y = 3.045x - 0.008$, $p < 0.001$, respectively). **b** H:N molar increases are significantly associated with O:C ratios following exposure to NH$_3$ ($y = 6.23x - 1.70$, $R^2 = 0.59$, $p < 0.005$, $F_{1,10} = 14.45$, $RSE_{10} = 0.53$), but not NH$_4^+$. PyOM samples retain <0.5 moles of NH$_3$–H for every mole of NH$_3$–N retained, compared to 2.13–23.60 moles of NH$_4^+$–H for every mole of NH$_4^+$–N retained. Green symbols represent original PyOM samples without N addition, orange symbols represent PyOM following exposure to NH$_4^+$, and blue symbols represent PyOM following exposure to NH$_3$. Shaded bands represent the 95% confidence intervals

**Fig. 1** Changes in pyrogenic organic matter ammonia retention and physiochemical characteristics as a function of molar O:C ratio. **a** ammonia (NH$_3$) retention capacity—expressed in mmol of NH$_3$ g$^{-1}$ of pyrogenic organic matter-carbon (PyOM–C, left $y$ axis) and percent nitrogen (N) of PyOM–C (right $y$ axis)—increases as a function of molar O:C ratio. Each point represents the average oxygen:carbon (O:C) ratio for two replicates. NH$_3$ chemisorption $= 17.49x - 1.84$, $R^2 = 0.89$, $p < 0.001$, $F_{1,25} = 204.7$, $S_{25} = 0.59$ (light blue); NH$_3$ combined chemical and physical adsorption $= 30.51x - 0.44$, $R^2 = 0.96$, $p < 0.001$, $F_{1,25} = 567.2$, $S_{25} = 0.62$ (dark blue). **b** Specific surface area (SSA) decreases as PyOM O:C ratio increases. SSA $= -365x + 288.8$, $R^2 = 0.931$, $p < 0.001$, $F_{1,25} = 338.5$, $S_{25} = 9.591$. **c** PyOM pH decreases as oxidation increases. Blue symbols represent unoxidized PyOM and PyOM incubated with deionized water (DIH$_2$O) and hydrogen peroxide (H$_2$O$_2$) and are fitted with a significant curve ($y = 20.8*e^{-14.8(O:C)} + 2.84$, $S_{19} = 0.199$). **d** The intensity of Fourier transform infrared (FTIR) peak heights associated with C=O stretching (1691–1715 cm$^{-1}$) increases in proportion to the intensity of peak heights associated with C=C vibrations and stretching (1582–1609 cm$^{-1}$) as PyOM O:C ratio increases ($y = 4.29x - 0.0670$, $R^2 = 0.963$, $p < 0.001$, $F_{1,25} = 650$, $S_{25} = 0.081$). For all figures, yellow symbols represent PyOM that was incubated with 1 M hydrochloric acid (HCl); pink symbols represent PyOM that was incubated with H$_2$O$_2$ and then with 1 M sodium hydroxide (NaOH); shaded bands represent the 95% confidence intervals

and exposure to oxidizing agents (e.g., microbial activity or ozone) when considering PyOM's potential role in N cycling.

**PyOM retains NH$_3$–N through chemisorption.** In addition to revealing PyOM's considerable NH$_3$ retention capacity, adsorption isotherms showed that up to 53% of the NH$_3$ was retained

through chemisorption rather than physisorption, and that this proportion was greatest in oxidized PyOM (Fig. 1a). A commonly proposed mechanism for NH$_3$ chemisorption by PyOM is protonation of NH$_3$ to form NH$_4^+$ and subsequent electrostatic interaction between the NH$_4^+$ and PyOM's negatively-charged functional groups[7], but our data indicate that this mechanism cannot solely be responsible for PyOM NH$_3$–N retention. Direct exposure of oxidized PyOM to NH$_4^+$ resulted in much lower N retention than exposure to NH$_3$ gas (Fig. 2a), suggesting that electrostatic interactions alone cannot explain PyOM's NH$_3$ retention capacity. Furthermore, if physisorption or electrostatic interactions are predominantly responsible for PyOM–N retention from NH$_3$ and NH$_4^+$, then stoichiometry dictates that on a molar basis, increases in PyOM–N following exposure should be accompanied by a threefold to fourfold increase in PyOM–H. However, when oxidized PyOM (molar O:C ratio 0.402) was exposed to NH$_3$, the increase in PyOM molar H:N ratio was smaller than 0.5, suggesting that a substantial portion of NH$_3$–N is retained without retention of NH$_3$–H (Fig. 2b). When the same PyOM sample was exposed to NH$_4^+$, the molar H:N ratio increased by 5.03, suggesting that most of the NH$_4^+$–N was retained along with NH$_4^+$'s H atoms. This stoichiometric comparison of H:N ratios in PyOM samples before and after exposure to NH$_3$ and NH$_4^+$ indicates that the respective mechanisms for N retention differ substantially, and that alternatives to physisorption and electrostatic interaction are likely responsible for PyOM's NH$_3$ retention capacity.

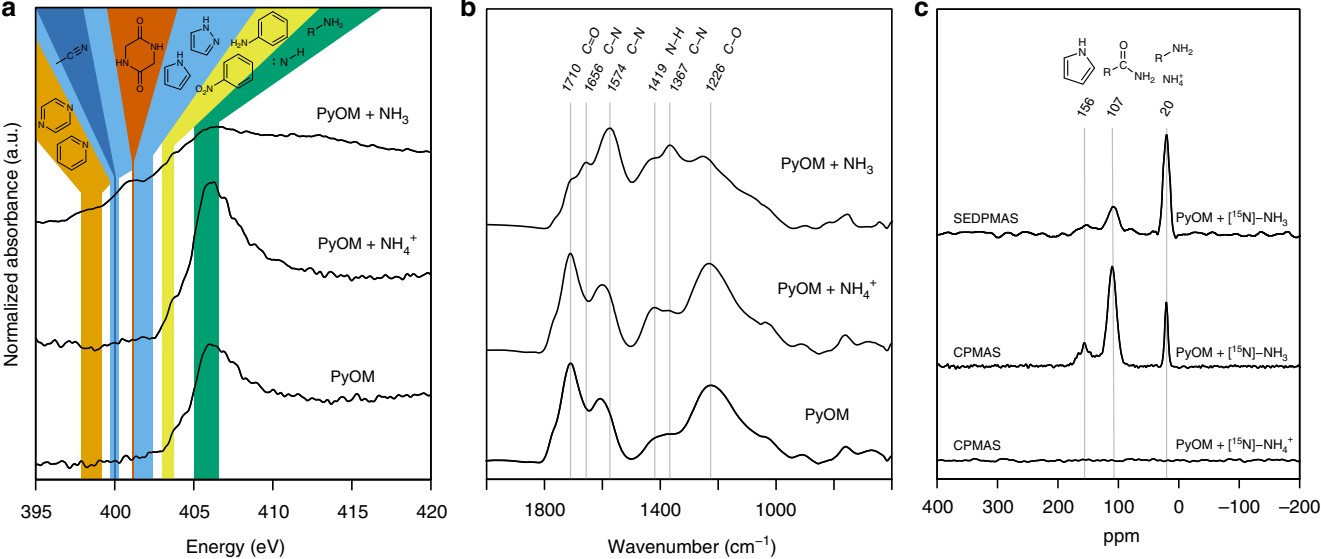

**Fig. 3** Nitrogen *K*-edge NEXAFS, FTIR, and NMR spectra of oxidized PyOM samples. Nitrogen (N) *K*-edge near-edge X-ray absorption fine structure (NEXAFS) (**a**), Fourier transform infrared (FTIR) (**b**), and nuclear magnetic resonance (NMR) (**c**) spectra collected from oxidized PyOM, oxidized PyOM following exposure to ammonium ($NH_4^+$), and oxidized pyrogenic organic matter (PyOM) following exposure to ammonia ($NH_3$). **a** Shaded bands represent the range of peak centers consistent with selected spectral features: 397.88–399.20 eV for C=N bonds in 1N and 2N aromatic six-membered rings (orange), 400.00 for nitrile bonds (dark blue), 399.76–400.27 for C=N bonds in 2N five-membered rings (light blue), 401.15 for C–N bonds in non-aromatic six-membered rings (red), 401.20–402.40 for C–N bonds in 1N and 2N aromatic five-membered rings (light blue), 403.00–403.75 for aliphatic N bonded to aromatic rings (yellow), and 405.00–406.58 for aliphatic amines and N–H bonds (green). Model chemical structures are shown at the top of the figure. **b** FTIR spectra of oxidized PyOM, oxidized PyOM following exposure to $NH_4^+$, and oxidized PyOM following exposure to $NH_3$. **c** $^{15}$N-NMR spin echo direct polarization magic angle spinning (SEDPMAS) spectrum of oxidized PyOM following exposure to [$^{15}$N]-NH$_3$, and $^{15}$N-NMR cross-polarization magic angle spinning (CPMAS) spectra of oxidized PyOM following exposure to [$^{15}$N]-NH$_3$ and [$^{15}$N]-NH$_4^+$. The spectra suggest that NH$_3$–N retention mechanisms could include $NH_4^+$ adsorption (represented by the peak at 20 ppm), and the formation of covalent C–N bonds such as amines (20 ppm), amides (~107 ppm), and aromatic five-membered heterocycles (chemical shifts between ~130–165 ppm)[16,33,39,40]

**PyOM and NH$_3$–N form covalent bonds**. To examine these alternative mechanisms for NH$_3$ retention, we compared the N near-edge X-ray absorption fine structure (*K*-edge NEXAFS) spectra of oxidized PyOM (O:C ratio 0.402) to those of oxidized PyOM following exposure to either NH$_3$ or NH$_4^+$ (Fig. 3a and Supplementary Fig. 5). This method cannot be used to quantify the absolute amount of N retained by PyOM, but does provide information about the types of covalent N bonds present. Exposure to NH$_3$ resulted in the formation of a variety of covalent N bonds that differ from those originating from PyOM feedstock N, N structures formed during thermal decomposition, N structures formed between PyOM and NH$_4^+$, and N–H bonds in pure NH$_3$ or NH$_4^+$[33–36]. Interactions between oxidized PyOM and NH$_3$ led to the strong development of absorption peaks between 397.88 and 402.40 eV, many of which are consistent with aromatic and non-aromatic heterocyclic N structures[14,36] (see Supplementary Tables 1–3). Compared to spectra collected from PyOM that was not exposed to additional N, spectra collected from PyOM following exposure to NH$_3$ showed a threefold increase in the $1s \rightarrow \pi^*$ area consistent with aromatic six-membered heterocycles containing either one or two N atoms (model peaks located at 397.88, 398.76, and 399.2 eV), a threefold increase in the area consistent with nitrile bonds and aromatic five-membered heterocycles containing either one or two N atoms (400.05, 401.43, and 402.40 eV), and a 1.3-fold increase in the area consistent with aliphatic N bonded to aromatic rings (403.00 and 403.65 eV) (Supplementary Table 2). Additionally, while no feature consistent with non-aromatic six-membered N heterocycles (401.15 eV) was identified in spectra collected from oxidized PyOM not exposed to NH$_3$, this feature accounted for two percent of the area underneath the spectrum collected from oxidized PyOM following exposure to NH$_3$. In contrast, development of

peaks in the $1s \rightarrow \pi^*$ region following exposure to NH$_4^+$ was very small, suggesting that there was little change in N functional group composition with NH$_4^+$ addition. Both the spectra of PyOM exposed to NH$_4^+$ as well as those of unexposed PyOM are strongly dominated by $1s \rightarrow \sigma^*$ features with peak centers between 405.00 and 406.58 eV, which dwarf the $1s \rightarrow \pi^*$ region area in these spectra. Although these $1s \rightarrow \sigma^*$ features are often associated with N–H bonds, they cannot be assigned definitively due to severe overlap of spectral features in this region[36].

To further compare mechanisms for PyOM retention of NH$_3$ in comparison to NH$_4^+$, we also collected FTIR spectra from oxidized PyOM, and oxidized PyOM following exposure to NH$_3$ and NH$_4^+$ (Fig. 3b). Similar to the NEXAFS spectra (Fig. 3a), the FTIR spectra show clear differences between functional groups present in PyOM, PyOM following exposure to NH$_3$, and PyOM following exposure to NH$_4^+$. Exposure to NH$_3$ resulted in the emergence of new peaks at 1656 and 1367 cm$^{-1}$ and an increase in peak height at 1574 cm$^{-1}$, all of which are consistent with C–N stretching, including C–N resonance stretching in aromatic rings at 1656 cm$^{-1}$[37,38]. Exposure to NH$_3$ also resulted in a marked decrease in the peaks associated with C=O and C–O carbonyl/carboxyl and ketonic stretching at 1710 and 1226 cm$^{-1}$, respectively, suggesting that these functional groups decrease relative to other functional groups in this sample. In contrast, exposure to NH$_4^+$ resulted in only two new spectral features, medium-sized peaks at 1419 and 1372 cm$^{-1}$, which are consistent with N–H and C–N stretching, respectively.

Definitive functional group assignment using FTIR spectra collected from heterogeneous materials such as PyOM is challenging because the regions associated with different bonds often overlap with one another. However, the major treatment difference between our PyOM samples is whether or not they

were exposed to gaseous $NH_3$ or aqueous $NH_4^+$. Therefore, although FTIR spectral features between 1200 and 1700 $cm^{-1}$ are sometimes associated with bonds between other elements (including C and O), it is probable that the emergence of distinct features in the FTIR spectra collected from our oxidized PyOM samples following exposure to either $NH_3$ or $NH_4^+$ is a result of bonds that formed between PyOM and N. Since the FTIR spectrum of pure $NH_3$ contains predominant peaks around 950 $cm^{-1}$, it is also unlikely that physical adsorption of $NH_3$ alone could account for the differences observed between the FTIR spectra collected from our PyOM samples before and after $NH_3$ exposure[38]. In contrast, the FTIR spectra of $NH_4^+$ standards contain predominant peaks around 1440 $cm^{-1}$, indicating that $NH_4^+$ adsorption was responsible for the relative increase in this region of the spectra collected from our PyOM samples following exposure to $NH_4^+$. This is consistent with our NEXAFS deconvolution analysis, which shows that exposure to $NH_4^+$ does not result in the substantial formation of a variety of N functional groups, despite the increased N content relative to unexposed PyOM samples. It also is consistent with elemental analyses, which suggest that oxidized PyOM retains $NH_4^+$–N in stoichiometric balance with $NH_4^+$–H, but retains $NH_3$–N without $NH_3$–H.

To further investigate mechanisms for $NH_3$–N and $NH_4^+$–N retention, we collected solid-state $^{15}$N-NMR spectra after separate exposure of oxidized PyOM to enriched $[^{15}N]$-$NH_3$ gas or $[^{15}N]$-$NH_4^+$ solution (Fig. 3c). Use of $^{15}$N-enriched reagents was necessary because there was insufficient signal from $^{15}$N at natural abundance in PyOM samples exposed to unlabeled $NH_3$ or $NH_4^+$. Substantial differences were observed in the $^{15}$N-NMR CPMAS spectra collected from PyOM exposed to enriched $[^{15}N]$-$NH_3$ gas and these differences confirmed the formation of a variety of new N functional groups, including $NH_4^+$ and amines (~20 ppm), and C–N groups such as amides (~107 ppm) and N heterocycles (~156 ppm)[16,33,39,40]. Similar to the results of our NEXAFS spectral deconvolution analyses, integration of the $^{15}$N-NMR SEDPMAS spectrum collected from oxidized PyOM following exposure to $[^{15}N]$-$NH_3$ gas shows that over 40% of the newly-incorporated NMR-detectable N is consistent with covalent C–N bonds, including more than 11% in heterocyclic structures (Supplementary Table 4). On the other hand, the $^{15}$N-NMR spectrum collected from PyOM exposed to $[^{15}N]$-$NH_4^+$ did not show any evidence of $NH_4^+$–N incorporation into PyOM, also corresponding with the results of our NEXAFS spectral deconvolution analysis, which show very little difference between the N functional group composition of PyOM and PyOM following exposure to $NH_4^+$. It is possible that some $NH_4^+$–N was incorporated into PyOM, but that the quantity retained was below the detection limit for NMR, implying that it is essential for an acid-base reaction (e.g., $-CO_2H + NH_3 \rightarrow -CO_2-NH_4^+$) to occur for $NH_4^+$ retention.

Direct comparison of NEXAFS and NMR results is difficult because of fundamental differences between the two methods. In particular, NMR detects functional groups, while NEXAFS detects individual bonds, some of which may be present together in one functional group. Additionally, due to severe overlap of features in the $1s \rightarrow \sigma^*$ region of NEXAFS N spectra, the portion of N–H and C–NH$_2$ bonds present in a sample cannot be determined with this method. However, the overall results from the two analyses are consistent. In combination with stoichiometric analyses, the NEXAFS, NMR, and FTIR spectra show that PyOM interactions with $NH_3$ under ambient conditions can result in substantial N retention and are fundamentally different from interactions with $NH_4^+$. The mechanism may involve nucleophilic $NH_3$ reacting with predisposed functional groups of PyOM, such as acid anhydrides, or diketo-fragments to form a range of

covalent C–N bonds, including amides and N heterocycles (Fig. 3). Similar reactions have been described between $NH_3$/$NH_4^+$ and small organic molecules such as carbonyls, glyoxals, and secondary organic aerosols found in the atmosphere[23,24,41]. Both the $NH_3$ retention capacity and mechanisms of natural PyOM are similar to those of some industrially produced graphene oxides and activated carbons[8,19,42] even when exposure occurs at ambient temperature and pressure. These results demonstrate for the first time that the enrichment of PyOM with N functional groups such as N heterocycles, aromatic N heterocycles, and amides may occur under natural environmental conditions and that PyOM's interaction with $NH_3$ versus $NH_4^+$ has very different implications for the global N cycle.

## Discussion

Our data show that natural PyOM—a ubiquitous component of soil, atmosphere, and water—can react with $NH_3$ gas to form covalent bonds under conditions approximating the natural environment. This is decisive because such covalent bond formation would result in more persistent N retention than physisorption, electrostatic interactions, and precipitation of $NH_4^+$ salts, which are currently thought to be the dominant mechanisms for PyOM $NH_3$ retention[7–9]. Since covalently bound N might be less accessible to living organisms and less susceptible to volatilization, diffusion, and leaching than weakly sorbed $NH_3$ and $NH_4^+$, it would also have very different implications for local N availability and global N cycling. By incorporating $NH_3$–N into covalent C–N bonds, PyOM could provide a more dynamic mechanism for N storage, transport, and release. The discovery of aromatic and non-aromatic heterocyclic N bond formation between natural PyOM and $NH_3$ at ambient temperatures is also noteworthy, as this has not been observed for terrestrial pyrogenic C material, including coal, activated carbon, and graphene oxides. This is particularly relevant for industrial applications, where N-doping is used to improve the performance of C-based supercapacitors, catalysts, and other materials[15,16,25].

The formation of covalent bonds between PyOM and $NH_3$–N under ambient conditions is a surprising outcome that has not been considered by most scientists investigating PyOM interactions with N. While the work presented here did not determine the chemical reactions responsible for such bond formation, similar reactions are well documented. For example, the reaction between carboxylic acids and amines (including $NH_3$ as the simplest case) to form amides is the basis of protein synthesis from amino acids. It is also well established that subsequent condensation, cyclization, and aromatization to form N-aromatics are also possible. For example, the Paal–Knorr pyrrole synthesis reaction—which produces pyrroles through the condensation of a dicarbonyl compound with an amine or $NH_3$—is thought to be responsible for N heterocycle formation between secondary organic aerosols and $NH_3$ or amines in the atmosphere[23,24,41]. The discovery of covalent bond formation between PyOM and $NH_3$ under ambient conditions may direct us to rethink PyOM material science and N biogeochemistry on local and global scales. Since many forms of organic matter present in the Earth's soil, atmosphere, and water contain the same functional groups found in PyOM, it is possible that similar reactions occur between these materials and $NH_3$. Future research should investigate the extent to which organic matter retains $NH_3$–N through covalent bonds, the mechanisms responsible, and the implications for global N biogeochemistry.

Given the existing uncertainties in global PyOM and $NH_3$ budgets, it is difficult to calculate exactly how much N might potentially be retained or transported by the Earth's PyOM stocks. Based on estimates of 54–109 Pg PyOM–C in soil[1] and an

$NH_3$ adsorption capacity of 13.5 mmol $g^{-1}$ for oxidized PyOM (Fig. 1a), we calculate that soil PyOM stocks have the potential to store or transport up to $7.3–14.7 \times 10^{14}$ mol $NH_3$ through PyOM–$NH_3$ interactions, equaling up to 645-fold more than estimated annual $NH_3$ emissions from global agriculture, or up to 251-fold more than the estimated quantity of annually applied synthetic N fertilizer[43]. If $NH_3$ interactions with soil PyOM are representative of those with other PyOM stocks, the atmospheric, ocean sediment, and marine PyOM pools could store or transport an additional $214 \times 10^{14}$ mol $NH_3$–N through similar mechanisms. Combined, all of these PyOM stocks could retain ~320 Pg N, or more than 1500-fold the contribution of global anthropogenic N inputs per year[44].

These calculations predict a large potential influence of PyOM on global N cycling, and should motivate further work to constrain estimates so that they reflect the amount of $NH_3$–N retained and transported by PyOM. It is important to consider factors influencing $NH_3$ volatilization (e.g., pH, moisture, and temperature), PyOM–$NH_3$ retention capacity (e.g., functional group composition, surface area, and fouling of PyOM surfaces), and other variables that affect interactions between PyOM and $NH_3$ (e.g., the temperature of exposure, distance from the $NH_3$ source, and biological competition for $NH_3$). However, even at relatively low $NH_3$ concentrations or PyOM–N retention levels, PyOM could influence $NH_3$ loss, N availability to plants and microbes, and global N transport. Additional experiments are necessary to investigate the frequency of PyOM–$NH_3$ interactions and to examine them in more complex and heterogeneous environments, especially in marine waters and sediments, which hold the vast majority of the Earth's PyOM stocks. The coupling of global C and N cycles through such interactions could also be significant and warrants further research, particularly as global fire patterns change.

## Methods

**PyOM preparation.** Maple (*Acer rubrum*) wood chips were pyrolyzed at 500 °C for 30 min in a modified muffle furnace[28]. In order to produce a homogenous product, the furnace employs a custom-made inline mixing unit, regulates temperature, and maintains an internal atmosphere of inert gas throughout pyrolysis. These highly standardized process conditions ensure that the pyrolysis products are as homogenous as possible. The resulting PyOM was ground and sieved to 149–850 μm, divided into subsamples, and incubated with hydrogen peroxide ($H_2O_2$) or deionized water ($DIH_2O$) at 30 °C for up to three months (PyOM:$H_2O_2$ ratio of 1:10 g $mL^{-1}$). After oxidation, PyOM was rinsed thoroughly with $DIH_2O$ and dried. Some PyOM samples were rewetted with $DIH_2O$ (PyOM: $DIH_2O$ ratio of 1:20 g $mL^{-1}$) and treated with 1 M HCl or NaOH until the desired pH was achieved. These PyOM samples were also rinsed with $DIH_2O$ and dried.

**PyOM characterization.** PyOM pH was measured in $DIH_2O$ at a ratio of 1:20 g $mL^{-1}$. SSA was quantified using the B.E.T. method with $CO_2$ at 273.15 K (ASAP 2020, Micromeritics, Atlanta, Georgia). Total C, N, H, and O were measured using a Delta V Isotope Ratio Mass Spectrometer (Thermo Scientific, Germany) coupled to a Carlo Erba NC2500 Elemental Analyzer (Italy).

**$NH_3$ and $NH_4^+$ adsorption.** $NH_3$ adsorption isotherms were measured with an Autosorb iQ gas sorption analyzer (Quantachrome Instruments, Boynton Beach, Florida). Briefly, samples were degassed at 300 °C for 3 h prior to $NH_3$ adsorption isotherm determination, which was conducted from 80 to 800 Torr at 35 °C. Chemisorption values indicate $NH_3$ that was retained by PyOM under vacuum. See Supplementary Fig. 6 for comparison of N measured by chemisorption isotherms to N measured by dry combustion. For $NH_4^+$- adsorption measurements, PyOM samples were mixed with 100 mM ammonium chloride solution for 16 h, filtered, rinsed with ethanol, and dried. Retained N was measured using an elemental analyzer, as described above.

**FTIR.** FTIR spectroscopy was used to characterize oxidized PyOM samples and investigate changes in PyOM functional group composition after exposure to $NH_3$ and $NH_4^+$. Two replicates of each PyOM sample were scanned 200 times from 575 to 3500 $cm^{-1}$ at a resolution of 8 $cm^{-1}$ using a Bruker Hyperion FT-IR Spectrometer (Bruker, Billerica, Massachusetts) equipped with a ZnSe crystal source (PIKE Technologies, Inc., Madison, Wisconsin). Atmospheric background spectra were

subtracted from each sample spectrum. Replicate sample spectra were averaged, baseline corrected, and normalized. Wavenumbers were assigned and peak ratios were calculated for the following functional groups: 752–761, 813–823, and 875–915 $cm^{-1}$ to aromatic C–H out of plane deformation, 1690–1715 $cm^{-1}$ to carbonyl/carboxyl and ketonic C=O stretching, and 1581–1609 $cm^{-1}$ to aromatic C=C vibrations and stretching (OPUS, Bruker, Billerica, Massachusetts).

**NEXAFS.** Nitrogen $K$-edge NEXAFS was used to discern how $NH_3$ and $NH_4^+$ were retained by PyOM following exposure. Briefly, samples were mounted onto gold-coated silicon wafers and scanned in 49 different locations for 20 seconds each, without any spatial overlap to prevent radiation damage to the sample. N $K_\alpha$ partial fluorescence yield was collected using silicon drift detectors in the slew scanning mode of the spherical grating monochromator beamline at the Canadian Light Source (Saskatoon, Canada). For each sample, all 49 scans were averaged across four detectors and normalized by the beamline incident flux obtained by measuring the drain current in a gold mesh (IGOR Pro 6.36, WaveMetrics, Lake Oswego, Oregon). Following a modification of the method used by Gillespie et al.[45], spectra were shifted based on the $N_2$ absorption spectrum measured from ammonium sulfate, background corrected, smoothed, and normalized to an edge-step of 1 (Athena 0.8.056, Bruce Ravel; Ifeffit 1.2.11, Matt Newville, University of Chicago, Chicago, Illinois). Deconvolution was performed using Gaussian curves and peak characteristics of N-containing standards (Fityk 0.9.8, Marcin Wojdyr; see Supplementary Fig. 7 for N standard spectra, Supplementary Table 1 for peak assignments used in deconvolution, and Supplementary Table 3 for features in spectra collected from standard compounds). The fraction of $\pi^*$ area associated with specific N bonds (compared to total area of all deconvolution products) was calculated for each sample (Supplementary Table 2). If they were present as physically adsorbed molecules, neither $NH_3$ nor $NH_4^+$ could have been responsible for the development of the numerous pre-edge features in the spectra collected from PyOM that was exposed to $NH_3$ and $NH_4^+$ (see Supplementary Figs. 7 and 8, Supplementary Table 2, and refs. [34–38]).

To confirm that radiation damage was not responsible for spectral features, samples were also scanned 15 additional times in the same location. These 15 spectra were then averaged, shifted, background corrected, smoothed, and normalized as described above. If the samples were susceptible to beam damage, we would expect to see new spectral features that would become more pronounced as each additional scan exposed the sample to increasing radiation. However, as shown in Supplementary Fig. 9, this did not occur—even after a 15-fold increase in radiation, the spectral features remain the same as those presented in Fig. 3a. As other authors have noted, this indicates that radiation damage was not responsible for the features present in NEXAFS sample spectra[46].

**Solid-state NMR spectroscopy.** Solid-state NMR spectroscopy was used to investigate how $NH_3$ and $NH_4^+$ were retained by oxidized PyOM following exposure. In order to obtain a sufficiently strong signal during $^{15}N$-NMR experiments, oxidized PyOM samples were exposed to gaseous $[^{15}N]$-$NH_3$ with 98 atom% $^{15}N$ (Air Liquide America Specialty Gases, Plumsteadville, PA) and $[^{15}N]$-$NH_4^+$ with 10 atom% $^{15}N$ (Cambridge Isotope Labs, Tewksbury, MA) at 35 °C, under atmospheric pressure.

1D $^1H$, $^{13}C$ and $^{15}N$ solid-state NMR spectra were obtained at a magnetic field of 7 Tesla ($^1H$, $^{13}C$ and $^{15}N$ Larmor frequency of 300 MHz, 75 and 30 MHz, respectively) using a Bruker Avance III NMR spectrometer fitted with a 4 mm magic angle spinning (MAS) double resonance probe. For both the $^{13}C$ and $^{15}N$ experiments, ~50 mg of PyOM was packed into a 4 mm zirconia rotor sealed with a Kel-F cap. For $^{13}C$ experiments, CP (cross-polarization) was achieved with 6.5 kHz MAS; contact time, 1 ms, ramped from 70 to 100%; recycle delay, 3–20 s (Supplementary Figs. 10 and 11); 83 kHz $^1H$ decoupling via spinal-64 sequence; ca 2k scans; TOSS (TOtal Suppression of Spinning side-bands) removed side-bands from the aromatic peaks that obscured the aliphatic region, and also gave better quality response from the sample without pulse breakthrough (Supplementary Figs. 4 and 12); SEDP (spin echo direct polarization with $^1H$ decoupling) was achieved with 12 kHz MAS; recycle delay of 2–150 s; $^{13}C$ excitation pulse of 4.5 μs (90°); echo time of ~75 μs; 71 kHz $^1H$ decoupling via spinal-64 sequence (Supplementary Fig. 13). Complementary spectra were also acquired with dipolar-dephasing delays of 40 μs with both CP and SEDP to assess non-protonated carbon content. For $^{15}N$ experiments, CP was achieved at 5 kHz MAS; contact time, 2 ms, ramped from 70 to 100%; recycle delay, 3 s; 83 kHz $^1H$ decoupling via spinal-64 sequence[47]. TOSS was also used to suppress pulse breakthrough. For $^{15}N$ SEDPMAS experiments, 10 kHz MAS was used with 100 μs echo time and a 50 kHz $^1H$ decoupling field during acquisition with the spinal-64 sequence. The relaxation behavior was tested using a series of experiments with a fixed number of scans (400) and with increasing recycle delays from 5 to 400 s (Supplementary Figs. 14 and 15).

All spectra were processed using the Bruker software, TOPSPIN 3.5pl7. Spectra were produced from the free induction decays by first zero filling, applying Gaussian multiplication (e.g., LB = −10, GB = 0.03), Fourier transformation, and phase correction. Chemical shift values were referenced to the C=O of glycine, $\delta_C$ 176 ppm for $^{13}C$ and to $(NH_4)_2SO_4$, $\delta_N$ 24 ppm on the $NH_3$ scale for $^{15}N$. All literature quoted in the text were converted to the $NH_3$ scale for $^{15}N$ by adding 380 ppm.

**Data analysis**. All statistical analyses were performed using the lsmeans and nlstools packages[48,49] in the statistical computing language and environment R[50].

## Data availability

The data that support the findings of this study are available in Cornell University's digital repository eCommons with the identifier https://doi.org/10.7298/X0B7-PX55.

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

## Acknowledgements

This work was supported by the National Science Foundation's BREAD Program (IOS-0965336), Cornell University's David R. Atkinson Center for a Sustainable Future, the Towards Sustainability Foundation, the Atkinson Center for a Sustainable Future Impact through Innovation Fund, USDA Hatch (NYC-125443), and the McKnight Foundation. Some of the research described in this paper was performed at the Canadian Light Source Inc., which is supported by the Natural Sciences and Engineering Research Council of Canada, the National Research Council Canada, the Canadian Institutes of Health

Research, the Province of Saskatchewan, Western Economic Diversification Canada, and the University of Saskatchewan. This work also made use of the Cornell Center for Materials Research Shared Facilities which are supported through the NSF MRSEC program (DMR-1120296, DMR-1719875). The Mark Wainwright Analytical Centre at the University of New South Wales is acknowledged for access to solid-state NMR spectrometers funded through Australian Research Council LIEF LE0989541. R.H. acknowledges support from the NSF IGERT Program (DGE-0903371 and DGE-1069193) and the NSF GRFP (DGE-1144153). Any opinions, findings, and conclusions or recommendations expressed in this material are those of the author(s) and do not necessarily reflect the views of the donors. Special thanks to Akio Enders, Kelly Hanley, and Cornell University Stable Isotope Laboratory staff for their help with sample analysis.

## Author contributions

R.H. and J.L. conceived the experiments; R.H. performed the experiments and analyzed the data; D.T.-R., J.D., A.G. and T.R. assisted with NEXAFS measurements and interpretation; J.H. conducted the NMR experiments; J.H. and R.S. led NMR data interpretation; R.H. wrote the paper; all authors contributed to the final draft.

## Additional information

**Competing interests:** The authors declare no competing interests.

