## [Peer Review File · Nature Communications]

Reviewers' comments:

Reviewer #1 (Remarks to the Author):

This manuscript examines pyrogenic organic matter (PyOM) interactions with ammonia and the chemisorption processes responsible for this. This builds on earlier work that identified PyOM had the capacity to take up ammonia but which did not identify mechanisms. Identifying covalent bonding mechanisms to be the dominant mechanism.

This study makes a significant and novel step towards increasing our understanding of PyOM-ammonia interactions. I question the significance of the finding in terms PyOM being able to retain 600-fold the annual NH₃ emissions from agriculture is correct. Yes it will be important but the pools of PyOM listed in soil, atmosphere, marine sediments, and ocean waters are not all going to be receiving ammonia e.g. PyOM buried in marine sediments or in soils at depth, or in non agricultural soils at depth simply will not be exposed to the same levels of ammonia as PyOM near a soil surface or in the atmosphere. PyOM has to be at the site of NH₃ formation and given NH₃ predominantly forms NH₄⁺ at most environmental pH values (<7.0) the implications for alkaline NH₃ gas to react with PyOM may be limited. So maybe some recalculation of the true significance and impact of this process is called for at a passive scale. Conversely, the identification of these mechanisms provides an opportunity to be proactive with forms of PyOM to mop up excess ammonia in systems where it is being produced using low-energy intensive methods.

The study is very nicely done and presented I see no issues with methodology or statistics, and the conclusions are sound - with the exception of the comment above.

I note some minor points:

L7 "...ammonia (NH₃)..."

L8 "...nitrogen (N)..."

L9 "...with a higher N content..."

L11 "...NH₃ retention increases 4-fold, of the NH₃ retained..."

L13 "...that NH₃..."

L15 "...600-fold the annual NH₃ emissions..."

L18 "... , atmosphere,..."

L25 "...emissions, N leaching, N availability..."

L26 "...with NH₃, the..."

L27 "...and which constitute a large part of the global total reactive N pool"

L32 "...Proposed mechanisms of NH₃..."

L39 "...composition. Thus it is unknown whether..."

L41 "...the environment, or if cyclic..."

L54 "...6-fold..."

L154 "...prior to NH₃ adsorption..."

Reviewer #2 (Remarks to the Author):

With the submitted manuscript the authors suggest that higher adsorption of NH₃ in PyOM is due to chemisorption. They further suggest that this mechanism can have a considerable impact on the N cycling in soils and on the interrelationship of the C and N cycle.

However, since quite a while, the introduction of NH₃ into aromatic structures has been an issue because it was suggested to allow the production of a slow N-release fertilizer which has comparable chemical characteristics to humic material. The process described here resembles that already described by the group of Fischer and Katzur (EP1144342, EP99968302.2, and US Patent

6695892) and represents a two-step procedure, in which the aromatic compound is oxidized by a strong acid to increase the number of reactive sites by oxidative alteration of the rather hydrophobic parent material. In a second step, the oxidized sites can react with ammonia at high temperature and high pressure conditions. I have seen publication showing that the reaction was possible at 70{degree sign}C but I am astonished to see that 35{degree sign}C is enough. This would indicate that covalent binding of NH₃ does not need much activation energy, which is hard to believe. I have to admit that I also have problems to see how this works under vacuum. Anyway, if this mechanism would work, it should be possible to add NH₃ to aromatic structures in soils under ambient conditions and heterocyclic aromatic structures should accumulate without any problems - in particular in hot climatic zones. Well, we don't have any indication that this is the case. In contrary, we have indication that almost all of the soil N is bound in amides and if there are heterocyclics they derive from burnt N-rich organic residues. Reviewing a bit the literature, I realized that there the exceptional high NH₃ adsorption was mainly related to the amount of carboxylic groups. However, in the literature this was explained with hydrogen bonding without the need of covalent binding. That the adsorption did not work so well for NH₄ may be due to the different experimental conditions and to the specific properties of NH₃. The only possible indications for covalent binding come from the NEXAFS spectra.

However, here several issues have to be clarified to be convincing.

1. For the identification of the nature of the bound N, NEXAFS is used, but it is not clear if the difference between PyOM and PyOM +NH₃ could be due to the presence of the N-gas? I suggest to deliver a spectrum of the pure NH₃ gas, too. How would the spectrum change if NH₃ forms hydrogen-bonds with carboxylic groups?

2. I am also astonished that PyOM does not show signals in the range of five-membered N-rings, but mainly of N-H. Published ¹⁵N NMR spectra clearly demonstrate that pyrrole-type structures are the dominant N form in PyOM. On the other hand, maple wood chips don't contain much N, so I am wondering which N-H groups are represented by the signal at 406.

Discussion:

1. The authors assume that the proposed reactions represent common reactions in soils. However, there are no reports - at least not to my knowledge - which could support this. Therefore, the assumption or suggestion proposed here, is highly speculative and I suggest to perform additional experiments which could support this conclusion. A possible experiment could be a comparable approach performed with soils mixed with PyOM or soils containing already aged - thus oxidized - PyOM.

2. Newer studies show that in soils PyOM is not as stable as assumed. Therefore, as soon as it will be oxidized its accessibility to soil organisms will increase, too. This will be in strong competition with chemical N fixation.

3. Several studies indicated that N-heterocyclic aromatic structures exhibit a low biochemical recalcitrance, thus they disappear rather quickly in aerated soils.(Fetzner 1998, de la Rosa and Knicker 2011, Lopez-Martin, Velasco-Molina et al. 2016).

Based on those considerations, I have to admit that I cannot recommend this work for publication in Nature Communications

de la Rosa, J. M. and H. Knicker (2011). "Bioavailability of N released from N-rich pyrogenic organic matter: An incubation study." *Soil Biology and Biochemistry* 43(12): 2368-2373.

Fetzner, S. (1998). "Bacterial degradation of pyridine, indole, quinoline, and their derivatives under different redox conditions." *Applied Microbiology and Biotechnology* 49(3): 237-250.

Lopez-Martin, M., M. Velasco-Molina and H. Knicker (2016). "Variability of the quality and quantity of organic matter in soil affected by multiple wildfires." *Journal of Soils and Sediments* 16(2): 360-370.

Reviewer #3 (Remarks to the Author):

This manuscript deals with the sorption of ammonia by pyrogenic organic matter (or charcoal). The manuscript is written well and the organization is adequate. Although interesting and the experimental data is collected over an extended period of time - the results/discussions are far from innovative [see below references].

The oldest reference cited in this article is 1994 - there were significant earlier advancements in the charcoal sorption literature, particular for charcoal to ammonia/ammonium. It is well established that charcoal can be activated with ammonium/ammonia gas (due to the chemical reactions occurring at the surface). Therefore, the current manuscript is lacking an innovative contribution and the calculations for the global N storage are not well developed given the fact that the data comes from solely a single hardwood charcoal - and no column leaching data, etc was used. The addition of a soil solution or dissolved DOM species is needed.

In addition, it has been known that charcoal interacts with water/oxygen/co₂ to form unique surface coverings -- it is inconclusive that these C-N bonds are formed from the sorbed CO₂ or actually the C originally contained in the PyOM.

Line 9 - express the complete units for the %

Line 120 - missing "x"

Garten, V. A., and D. E. Weiss. "The quinone-hydroquinone character of activated carbon and carbon black." *Australian Journal of Chemistry* 8, no. 1 (1955): 68-95.

Forney, William E. "Process for manufacturing nitrogen derivatives of carbon compounds." U.S. Patent 2,331,968, issued October 19, 1943.

Mortland, M. M. (1958). Reactions of ammonia in soils. *Adv. Agron*, 10, 325-348.

Richardson, Leon B. "THE ADSORPTION OF CARBON DIOXIDE AND AMMONIA BY CHARCOAL." *Journal of the American Chemical Society* 39, no. 9 (1917): 1828-1848.

Bancroft, W. D. (1919). "Charcoal before the War. III." *The Journal of Physical Chemistry* 24(5): 342-366.

Othmer, D. F. and F. G. Sawyer (1943). "Correlating Adsorption Data." *Industrial & Engineering Chemistry* 35(12): 1269-1276.

Hatch Jr, T. F., & Pigford, R. L. (1962). Simultaneous absorption of carbon dioxide and ammonia in water. *Industrial & Engineering Chemistry Fundamentals*, 1(3), 209-214.

Cacace, F., & Wolf, A. P. (1962). The Effect of Radiation on the Reactions of Recoil Carbon-II In Ammonia. *Journal of the American Chemical Society*, 84(16), 3202-3204.

Miller, R. S., D. Y. Curtin and I. C. Paul (1974). "Reactions of molecular crystals with gases. I. Reactions of solid aromatic carboxylic acids and related compounds with ammonia and amines." *Journal of the American Chemical Society* 96(20): 6329-6334.

Meredith, J. M. and C. A. Plank (1967). "Adsorption of carbon dioxide and nitrogen on charcoal at at 30.degree. to 50.degree." *Journal of Chemical & Engineering Data* 12(2): 259-261.

Puri, B. R., B. Kaistha, Y. Vardhan and O. Mahajan (1973). "Studies in surface chemistry of carbon blacks-vi. adsorption isotherms of benzene on carbons associated with different surface oxygen complexes." *Carbon* 11(4): 329-336.

Prober, R., J. J. Pyeha and W. E. Kidon (1975). "Interaction of activated carbon with dissolved

oxygen." *AIChE Journal* 21(6): 1200-1204.

Boehm, H-P., E. Diehl, W. Heck, and R. Sappok. "Surface oxides of carbon." *Angewandte Chemie International Edition in English* 3, no. 10 (1964): 669-677.

Stoeckli, H., F. Kraehenbuehl and D. Morel (1983). "The adsorption of water by active carbons, in relation to the enthalpy of immersion." *Carbon* 21(6): 589-591.

Shrier, A. L., and P. V. Danckwerts. "Carbon dioxide absorption into amine-promoted potash solutions." *Industrial & Engineering Chemistry Fundamentals* 8, no. 3 (1969): 415-423.

Thank you for your constructive comments on our manuscript. Please see our responses in red below.

Reviewer #1 (Remarks to the Author):

This manuscript examines pyrogenic organic matter (PyOM) interactions with ammonia and the chemisorption processes responsible for this. This builds on earlier work that identified PyOM had the capacity to take up ammonia but which did not identify mechanisms. Identifying covalent bonding mechanisms to be the dominant mechanism.

This study makes a significant and novel step towards increasing our understanding of PyOM-ammonia interactions. I question the significance of the finding in terms PyOM being able to retain 600-fold the annual NH₃ emissions from agriculture is correct. Yes it will be important but the pools of PyOM listed in soil, atmosphere, marine sediments, and ocean waters are not all going to be receiving ammonia e.g. PyOM buried in marine sediments or in soils at depth, or in non agricultural soils at depth simply will not be exposed to the same levels of ammonia as PyOM near a soil surface or in the atmosphere.

Response: This is a good point. We agree that buried PyOM might be exposed to different levels of NH₃ than PyOM in the surface or atmosphere. The 600-fold estimate is based on soil PyOM alone. If we extend this calculation to include PyOM contained in the atmosphere, sediments, and oceans, this estimate would increase another five- to fifteen-fold. Even if less than 1/1000th of soil PyOM retained NH₃, this could contribute to substantial retention of NH₃ emissions from fertilizers, natural decomposition of soil organic matter, and other sources. However, we did adjust the manuscript to reflect this uncertainty (see the last paragraph in the Discussion section).

PyOM has to be at the site of NH₃ formation and given NH₃ predominantly forms NH₄⁺ at most environmental pH values (<7.0) the implications for alkaline NH₃ gas to react with PyOM may be limited.

Response: We do not think that PyOM has to be at the site of NH₃ formation in order to interact with NH₃. In fact, this is one of the reasons that this interaction is so interesting— NH₃ gas can move from the emission source, through soil, atmosphere, and water, and then interact with PyOM that is located somewhere else. Regarding pH—it is true that below pH 7, NH₄⁺ will predominate. We have amended the manuscript to reflect this (see the last paragraph in the Discussion section).

So maybe some recalculation of the true significance and impact of this process is called for at a passive scale. Conversely, the identification of these mechanisms provides an opportunity to be proactive with forms of PyOM to mop up excess ammonia in systems where it is being produced using low-energy intensive methods.

The study is very nicely done and presented I see no issues with methodology or statistics, and the conclusions are sound - with the exception of the comment above.

I note some minor points:

L7 "...ammonia (NH₃)..."

L8 "...nitrogen (N)..."

L9 "...with a higher N content..."

L11 "...NH₃ retention increases 4-fold, of the NH₃ retained..."

L13 "...that NH₃..."

L15 "...600-fold the annual NH₃ emissions..."

L18 "..., atmosphere,..."

L25 "...emissions, N leaching, N availability..."

L26 "...with NH₃, the..."

L27 "...and which constitute a large part of the global total reactive N pool"

L32 "...Proposed mechanisms of NH₃..."

L39 "...composition. Thus it is unknown whether..."

L41 "...the environment, or if cyclic..."

L54 "...6-fold..."

L154 "...prior to NH₃ adsorption..."

Response: Thank you for catching these. We adopted all of your suggestions into the manuscript.

Reviewer #2 (Remarks to the Author):

With the submitted manuscript the authors suggest that higher adsorption of NH₃ in PyOM is due to chemisorption. They further suggest that this mechanism can have a considerable impact on the N cycling in soils and on the interrelationship of the C and N cycle.

However, since quite a while, the introduction of NH₃ into aromatic structures has been an issue because it was suggested to allow the production of a slow N-release fertilizer which has comparable chemical characteristics to humic material. The process described here resembles that already described by the group of Fischer and Katzur (EP1144342, EP99968302.2, and US Patent 6695892) and represents a two-step procedure, in which the aromatic compound is oxidized by a strong acid to increase the number of reactive sites by oxidative alteration of the rather hydrophobic parent material. In a second step, the oxidized sites can react with ammonia at high temperature and high pressure conditions.

Response: The processes described in these patents differ substantially from the process that we describe in our manuscript (see summary below). The authors of these patents use lignite instead of PyOM—two very different materials that have dissimilar chemical composition and geographical distribution. The authors also expose the lignite to NH₃ at temperatures reaching 100 C, which far exceeds ambient environmental temperatures. Furthermore, the authors do not show any evidence that covalent bonds form between NH₃ and lignite providing only anecdotal support, such as the claim that N forms contained in lignite “differ by their hydrolyzability.”

Summary: US Patent 6695892 states that amide bonds can form when lignite (also known as brown coal, made from compressed peat) is exposed to NH₃ at temperatures reaching 100 C. EP 1144342 states that amide, ammonium, “organically bonded,” and bonds “not hydrolysable as amide organically bonded” can form between lignite and NH₃ at temperatures reaching 100 C. EP 99968302.2 makes similar claims.

I have seen publication showing that the reaction was possible at 70{degree sign}C but I am astonished to see that 35{degree sign}C is enough. This would indicate that covalent binding of NH₃ does not need much activation energy, which is hard to believe. I have to admit that I also have problems to see how this works under vacuum.

Response: We were also surprised and excited to see that these reactions are possible at 35 C, a temperature that occurs naturally in ecosystems throughout the world. This comment supports our assertion of novelty.

Anyway, if this mechanism would work, it should be possible to add NH₃ to aromatic structures in soils under ambient conditions and heterocyclic aromatic structures should accumulate without any problems - in particular in hot climatic zones. Well, we don't have any indication that this is the case. In contrary, we have indication that almost all of the soil N is bound in amides and if there are heterocyclics they derive from burnt N-rich organic residues.

Response: We are not claiming that most soil N is bound in heterocyclic rings, but that NH₃ can interact with PyOM to form covalent bonds, including non-cyclic amides, heterocyclic rings, and other structures. Our experiments provide evidence that this is possible under laboratory conditions and indicate that it may also occur in the environment. We think that it is important to know this regardless of whether or not these N structures accumulate in soil, and whether or not most soil N is bound in amides. Additionally, multiple studies published over the past decade do show that cyclic N can accumulate in soils (Gillespie *et al.* 2014, Smernik & Baldock 2005). Although some of these cyclic N structures may be part of burnt residues, there are many other sources of cyclic N, including nucleic acids and amino acids, among others. However, because these points are beyond the goals and scope of our study, and we chose not to include them in our manuscript.

Reviewing a bit the literature, I realized that there the exceptional high NH₃ adsorption was mainly related to the amount of carboxylic groups. However, in the literature this was explained with hydrogen bonding without the need of covalent binding.

Response: We agree that NH₃ retention is positively correlated with carboxylic groups (see Figure 1 and Supplementary Figure 3 in our manuscript). It is possible that hydrogen bonding might contribute to NH₃ retention. However, occurrence of hydrogen bonding does not exclude the possibility of covalent bond formation between NH₃ and PyOM at ambient temperature, which is the novel aspect of our manuscript. Using NEXAFS, we found evidence of such covalent bond formation. Other authors have also observed covalent bonding after exposing industrially-produced activated carbon or graphene oxide to NH₃ at high temperatures that are not environmentally relevant (see papers cited in our manuscript, e.g., Schultz *et al.* 2014).

That the adsorption did not work so well for NH₄ may be due to the different experimental conditions and to the specific properties of NH₃.

Response: We fully agree that the specific properties of NH₃ compared to NH₄⁺ influence the way that they interact with PyOM (see the last part of the Results section). This is interesting because in previous papers studying PyOM-NH₃ interactions, authors have assumed that upon exposure, NH₃ is protonated into NH₄⁺ and retained through electrostatic attraction. Our results show that NH₃ behavior is fundamentally different from NH₄⁺, and that this has significant consequences for N retention.

The only possible indications for covalent binding come from the NEXAFS spectra.

Response: NEXAFS is an excellent way to investigate whether or not covalent bonding occurs. We considered using NMR in addition to NEXAFS, but since NMR does not accurately detect heterocyclic N in organic matter, we decided to employ the more recently developed NEXAFS spectroscopy, which utilizes the most advanced synchrotron system available (see Smernik & Baldock 2005; Leinweber *et al.* 2013).

However, here several issues have to be clarified to be convincing.

1. For the identification of the nature of the bound N, NEXAFS is used, but it is not clear if the difference between PyOM and PyOM +NH₃ could be due to the presence of the N-gas? I suggest to deliver a spectrum of the pure NH₃ gas, too. How would the spectrum change if NH₃ forms hydrogen-bonds with carboxylic groups?

Response: These are good questions. NEXAFS cannot be used to observe electrostatic bonds, such as hydrogen bonds. Therefore, although hydrogen bonding between NH₃ and carboxylic groups may be possible, it could not be responsible for the spectral features associated with 5- and 6-membered N rings, nitrile bonds, and N bonded to aromatic rings in our NEXAFS spectra. Similarly, NH₃ gas could not be responsible for these spectral features. The spectrum of NH₃ gas is distinctly different from our PyOM+NH₃ spectra and could not account for the multiple pre-edge features present in these spectra but not in the spectra collected from the unexposed PyOM or the PyOM+NH₄⁺ (see Jaeger *et al.* 1983 and Wight & Brion 1974, which show NH₃ spectra and are now cited in the manuscript; also see Schultz *et al.* 2014 and Geng *et al.* 2011, which attribute N K-edge NEXAFS spectral features to NH₃-N that has been incorporated into the graphene oxide structure rather than to NH₃ gas). We have added information to the NEXAFS methods and results sections to make this clearer.

2. I am also astonished that PyOM does not show signals in the range of five-membered N-rings, but mainly of N-H. Published ¹⁵N NMR spectra clearly demonstrate that pyrrole-type structures are the dominant N form in PyOM. On the other hand, maple wood chips don't contain much N, so I am wondering which N-H groups are represented by the signal at 406.

Response: Actually, deconvolution of the NEXAFS spectra did detect spectral features associated with 5-membered N rings in the PyOM NEXAFS spectra, but they are proportionally smaller those in the PyOM+NH₃ spectra (see Supplementary Table 2 for Gaussian curve areas associated with these N structures). The peak near 406 eV is associated with several N structures, including N-H bonds in 5-membered rings (see Supplementary Table 1 for peak assignments). Also, typically, NEXAFS spectra are normalized so that the pre- and post-edge

regions range from 0 to 1 (see the NEXAFS method section). Therefore, although the peak around 406 eV may seem large in the PyOM spectrum, this does not indicate that the PyOM is rich in N.

Discussion:

1. The authors assume that the proposed reactions represent common reactions in soils. However, there are no reports - at least not to my knowledge - which could support this. Therefore, the assumption or suggestion proposed here, is highly speculative and I suggest to perform additional experiments which could support this conclusion. A possible experiment could be a comparable approach performed with soils mixed with PyOM or soils containing already aged - thus oxidized - PyOM.

Response: We did not mean to suggest that this reaction is common, only that it is possible and therefore should inform our understanding of N cycling, since PyOM and NH₃ are both ubiquitous (see the last paragraph of the Discussion section). Although conducting an experiment with soil mixtures would be interesting, it would not allow us to gain the mechanistic insight that our study provides.

2. Newer studies show that in soils PyOM is not as stable as assumed. Therefore, as soon as it will be oxidized its accessibility to soil organisms will increase, too. This will be in strong competition with chemical N fixation.

Response: This may be true, but does not mean that the process we have described does not occur. More importantly, it is beyond the scope of our study, which was not intended to address the persistence of PyOM in soils (an entirely different, albeit very interesting topic). We would expect covalent bond formation between NH₃ and PyOM to occur on a faster timescale than microbial activity, but have amended the manuscript to address this possibility (see the last paragraph of the Discussion section).

3. Several studies indicated that N-heterocyclic aromatic structures exhibit a low biochemical recalcitrance, thus they disappear rather quickly in aerated soils. (Fetzner 1998, de la Rosa and Knicker 2011, Lopez-Martin, Velasco-Molina et al. 2016).

Response: The persistence of N heterocycles in soils is beyond the scope of our study. Our paper discusses the interactions that occur between PyOM and NH₃, not the persistence of PyOM in soils.

Based on those considerations, I have to admit that I cannot recommend this work for publication in Nature Communications

de la Rosa, J. M. and H. Knicker (2011). "Bioavailability of N released from N-rich pyrogenic organic matter: An incubation study." *Soil Biology and Biochemistry* 43(12): 2368-2373.
Fetzner, S. (1998). "Bacterial degradation of pyridine, indole, quinoline, and their derivatives under different redox conditions." *Applied Microbiology and Biotechnology* 49(3): 237-250.
Lopez-Martin, M., M. Velasco-Molina and H. Knicker (2016). "Variability of the quality and quantity of organic matter in soil affected by multiple wildfires." *Journal of Soils and Sediments* 16(2): 360-370.

Response: Thank you for highlighting these papers. We have added the paper by de la Rosa and Knicker to our reference list.

Reviewer #3 (Remarks to the Author):

This manuscript deals with the sorption of ammonia by pyrogenic organic matter (or charcoal). The manuscript is written well and the organization is adequate. Although interesting and the experimental data is collected over an extended period of time - the results/discussions are far from innovative [see below references].

The oldest reference cited in this article is 1994 - there were significant earlier advancements in the charcoal sorption literature, particular for charcoal to ammonia/ammonium.

Response: We acknowledge that research on charcoal predates 1994, but focused on the papers we thought were most pertinent to our manuscript. Our current draft includes an augmented reference list, including some of the references that you provided below.

It is well established that charcoal can be activated with ammonium/ammonia gas (due to the chemical reactions occurring at the surface).

Response: In this paper, we describe a different process, whereby PyOM is oxidized (as it would be when exposed to natural oxidizing forces in the environment) and then reacts with NH_3 . Although there are papers describing interactions between NH_3 and charcoal, activated carbon, and other industrial forms of pyrogenic carbon, none of these have observed the formation of cyclic N structures when these materials are exposed to NH_3 at temperatures that are environmentally relevant. Also, none have observed the formation of covalent bonds between natural PyOM and NH_3 . Therefore, although relevant previous studies exist, our work is the first to demonstrate that this process is possible in the natural environment.

Therefore, the current manuscript is lacking an innovative contribution and the calculations for the global N storage are not well developed given the fact that the data comes from solely a single hardwood charcoal - and no column leaching data, etc was used. The addition of a soil solution or dissolved DOM species is needed.

Response: We agree that it would be interesting to investigate whether PyOM- NH_3 interactions vary with different PyOM feedstocks. We chose wood because it is a very common component of global PyOM stocks. Although they would be interesting, we do not think that column leaching data or the addition of a soil solution or dissolved DOM would contribute to the main message of our study.

In addition, it has been known that charcoal interacts with water/oxygen/ CO_2 to form unique surface coverings -- it is inconclusive that these C-N bonds are formed from the sorbed CO_2 or actually the C originally contained in the PyOM.

Response: We agree that oxidation can influence the surface functional characteristics of PyOM (see Figure 1d and Supplementary Figures 1 and 3), although in our samples this resulted in increased carboxylic groups and reduced aromaticity more than in physisorbed CO_2 . Regardless of whether NH_3 interacts with PyOM alone or PyOM that has sorbed atmospheric CO_2 , this still results in the formation of covalent N structures (including heterocycles, nitriles, etc.) that can

be stored and transported by PyOM. Now that we have shown that this is possible under environmentally relevant conditions, we hope that additional research can address these questions.

Line 9 - express the complete units for the %

Response: We converted the units into mg g^{-1} .

Line 120 - missing "x"

Response: We added this to the manuscript.

Response: While interesting, none of the papers listed below show that covalent bond formation occurs when PyOM is exposed to NH_3 under ambient conditions, and several of the papers are only tangentially related to our work. A brief summary follows each citation. We added some of these papers to the list cited in our manuscript.

Garten, V. A., and D. E. Weiss. "The quinone-hydroquinone character of activated carbon and carbon black." *Australian Journal of Chemistry* 8, no. 1 (1955): 68-95.

Response: This paper describes the chemical structures of activated carbon and carbon blacks, neither of which is naturally-occurring.

Forney, William E. "Process for manufacturing nitrogen derivatives of carbon compounds." U.S. Patent 2,331,968, issued October 19, 1943.

Response: This patent describes the exposure of petroleum oil distillate to NH_3 at temperatures exceeding 400 C, a process that is not environmentally relevant. The author claims that this results in the formation of naphthylamines and alkyl amines (not heterocyclic structures), but it is not clear how this was assessed.

Mortland, M. M. (1958). Reactions of ammonia in soils. *Adv. Agron*, 10, 325-348.

Response: This paper describes NH_3 sorption to clay minerals and organic matter. This supports our findings that PyOM could interact with NH_3 in soil; we take this several steps further by providing evidence that this interaction might not be a simple adsorption, but actually results in covalent bond formation.

Richardson, Leon B. "THE ADSORPTION OF CARBON DIOXIDE AND AMMONIA BY CHARCOAL." *Journal of the American Chemical Society* 39, no. 9 (1917): 1828-1848.

Response: The author exposed charcoal to CO_2 and NH_3 at temperatures between -64 C and 200 C. He concludes that NH_3 adsorbs to charcoal. This conclusion is based on sorption isotherms—there is no data presented that could support or refute whether covalent bonds form between charcoal and NH_3 .

Bancroft, W. D. (1919). "Charcoal before the War. III." *The Journal of Physical Chemistry* 24(5): 342-366.

Response: The author mentions a few procedures involving charcoal, NH_3 , and various other substances, but the results are described qualitatively (e.g., "When the residual charcoal is washed with dilute hydrochloric acid and then with ammonia, large amounts are obtained of the brownish black colloid...when the diamond powder was washed with hydrochloric acid and then treated with aqueous ammonia, a heavy, white, milky suspension was obtained...").

Othmer, D. F. and F. G. Sawyer (1943). "Correlating Adsorption Data." *Industrial & Engineering Chemistry* 35(12): 1269-1276.

Response: The authors discuss the relationship between gas temperature, concentration, and vapor pressure to the gas's adsorption to activated carbon. The experiments do not use PyOM, nor do they investigate mechanisms for NH_3 retention.

Hatch Jr, T. F., & Pigford, R. L. (1962). Simultaneous absorption of carbon dioxide and ammonia in water. *Industrial & Engineering Chemistry Fundamentals*, 1(3), 209-214.

Response: The authors found that $(\text{NH}_4)_2\text{CO}_3$ formation was possible when CO_2 and NH_3 were absorbed in water. We know that NH_4^+ salt formation is one of several possible mechanisms for N retention by PyOM (see Day *et al.* 2005, cited in our manuscript). One of the most surprising and novel findings of our study is that covalent bond formation (including several cyclic structures) is also an N retention mechanism for PyOM exposed to NH_3 at ambient temperatures. Until now, authors discussing PyOM's interactions with NH_3 assume that physisorption, electrostatic interactions, and NH_4^+ salt formation are the only relevant N retention mechanisms.

Cacace, F., & Wolf, A. P. (1962). The Effect of Radiation on the Reactions of Recoil Carbon-II In Ammonia. *Journal of the American Chemical Society*, 84(16), 3202-3204.

Response: This article investigates the radioactive decay of N to C radio isotopes. It does not investigate the interaction between NH_3 and C.

Miller, R. S., D. Y. Curtin and I. C. Paul (1974). "Reactions of molecular crystals with gases. I. Reactions of solid aromatic carboxylic acids and related compounds with ammonia and amines." *Journal of the American Chemical Society* 96(20): 6329-6334.

Response: The authors found that carboxylic acids could react with NH_3 to form NH_4^+ salts (see comment above; in brief, this is the current knowledge that we juxtapose with our findings—that in fact, non-cyclic and heterocyclic N structures can also form).

Meredith, J. M. and C. A. Plank (1967). "Adsorption of carbon dioxide and nitrogen on charcoal at at 30.degree. to 50.degree." *Journal of Chemical & Engineering Data* 12(2): 259-261.

Response: The authors exposed charcoal to N_2 gas (not NH_3). This is a very different reaction than the one we've described in our manuscript.

Puri, B. R., B. Kaistha, Y. Vardhan and O. Mahajan (1973). "Studies in surface chemistry of carbon blacks-vi. adsorption isotherms of benzene on carbons associated with different surface oxygen complexes." *Carbon* 11(4): 329-336.

Response: The authors found that benzene can adsorb strongly to carbon blacks. It is not clear how this is related to covalent bond formation between PyOM and NH_3 .

Prober, R., J. J. Pyeha and W. E. Kidon (1975). "Interaction of activated carbon with dissolved oxygen." *AIChE Journal* 21(6): 1200-1204.

Response: The authors found that carboxylic acids increased when activated carbon was exposed to dissolved oxygen. No experiments with NH_3 are included.

Boehm, H-P., E. Diehl, W. Heck, and R. Sappok. "Surface oxides of carbon." *Angewandte Chemie International Edition in English* 3, no. 10 (1964): 669-677.

Response: While justifying our study of the effect of oxidation on PyOM's behavior, this paper does not investigate NH_3 .

Stoeckli, H., F. Kraehenbuehl and D. Morel (1983). "The adsorption of water by active carbons, in relation to the enthalpy of immersion." *Carbon* 21(6): 589-591.

Response: The authors study water adsorption to active carbon, a material that is not found in the environment. They conclude that these adsorption isotherms can be used to characterize the active carbon micropores. While interesting, it does not provide information about interactions between natural PyOM and NH_3 .

Shrier, A. L., and P. V. Danckwerts. "Carbon dioxide absorption into amine-promoted potash solutions." *Industrial & Engineering Chemistry Fundamentals* 8, no. 3 (1969): 415-423.

Response: The authors find that adding amines to potash solutions increases CO_2 adsorption into the solution. Unfortunately, there is no investigation with NH_3 included in this study.

Reviewers' comments:

Reviewer #1 (Remarks to the Author):

The authors have investigated pyrogenic organic matter (PyOM) reactions with ammonia showing ammonia retention under ambient conditions, and with ammonia retention altering with weathering. Further analysis demonstrates covalent bonding as a cause.

The manuscript is interesting and highly topical. I find the approach taken to be careful with appropriate statistical and quality checks on the study. As a consequence the conclusions are sound. The authors have responded to the original critiques of the reviewers carefully and rationally and demonstrated the originality of the current study.

I found no grammatical or typographic errors.

I think the authors have clearly presented some novel work. I recommend the study is published.

Reviewer #2 (Remarks to the Author):

Response of the authors: The processes described in these patents differ substantially from the process that we describe in our manuscript (see summary below). The authors of these patents use lignite instead of PyOM—two very different materials that have dissimilar chemical composition and geographical distribution. The authors also expose the lignite to NH₃ at temperatures reaching 100 C, which far exceeds ambient environmental temperatures. Furthermore, the authors do not show any evidence that covalent bonds form between NH₃ and lignite providing only anecdotal support, such as the claim that N forms contained in lignite “differ by their hydrolyzability.”

Summary: US Patent 6695892 states that amide bonds can form when lignite (also known as brown coal, made from compressed peat) is exposed to NH₃ at temperatures reaching 100 C. EP 1144342 states that amide, ammonium, “organically bonded,” and bonds “not hydrolysable as amide organically bonded” can form between lignite and NH₃ at temperatures reaching 100 C. EP 99968302.2 makes similar claims.

New comment of the reviewer:

Well, the difference between lignite and PyOM depends upon the source of PyOM. It is true that PyOM from wood produced during vegetation fires will contain a considerable amount of furan-type structures, whereas lignite contains more lignin derivatives. However, both contain aromatic structures

Response of the authors: We were also surprised and excited to see that these reactions are possible at 35 C, a temperature that occurs naturally in ecosystems throughout the world. This comment supports our assertion of novelty.

New comment of the reviewer: If I am honest, it does not support your assertion of novelty but sincere doubts. What are the activation energies for forming covalent bonds? How was this activation energy reduced?

Response of the authors: We are not claiming that most soil N is bound in heterocyclic rings, but that NH₃ can interact with PyOM to form covalent bonds, including non-cyclic amides, heterocyclic rings, and other structures. Our experiments provide evidence that this is possible under laboratory conditions and indicate that it may also occur in the environment. We think that it is important to know this regardless of whether or not these N structures accumulate in soil, and whether or not most soil N is bound in amides. Additionally, multiple studies published over the past decade do show that cyclic N can accumulate in soils (Gillespie et al. 2014, Smernik &

Baldock 2005). Although some of these cyclic N structures may be part of burnt residues, there are many other sources of cyclic N, including nucleic acids and amino acids, among others. However, because these points are beyond the goals and scope of our study, and we chose not to include them in our manuscript.

New Comments of the reviewer:

Other studies showed that heterocyclic N has low biochemical stability and there are a wide range of microorganisms which are able to degrade such compounds (i.e. Fetzner, Appl. Microbiol. Biotechnol. 49, 237-250, 1998). The contribution of Porphyrin from Chlorophyll is less than 1% of the dry mass of plants, the contribution of histidine to the total amino acid N in Grass is between 3 to 4% and that of DNA-N is not much higher.

Now, I was not saying that the authors are claiming that most soil N is bound in heterocyclic rings but if the proposed reaction occurs, there should be much more heterocyclic N in soils than presently identified. Thus the lack of higher amounts of heterocyclic N in unburnt soils points against the occurrence of the suggested reaction in natural soils.

Response of the author: NEXAFS is an excellent way to investigate whether or not covalent bonding occurs. We considered using NMR in addition to NEXAFS, but since NMR does not accurately detect heterocyclic N in organic matter, we decided to employ the more recently developed NEXAFS spectroscopy, which utilizes the most advanced synchrotron system available (see Smernik & Baldock 2005; Leinweber et al. 2013).

New comment of the reviewer: It is simply not true that NMR does not accurately detect heterocyclic N in organic matter and there is no physical reason why it should be a problem, if correct acquisition parameters are applied. In the literature there are enough examples showing that NMR can be used for that purpose and several publications show solid-state ^{15}N NMR spectra of burnt soils with a dominance of heterocyclic N. However, even if one does not trust in the quantification of such spectra, the formation of such compounds by the proposed reaction should be indicated by an increasing signal in the respective chemical shift region, in particular if the proposed reactions is of such high importance as claimed (a general rule is that C and N should be visible by CPMAS NMR if the distance is not higher than two to three bonds.....)

Reviewer #3 (Remarks to the Author):

The revised manuscript is greatly improved and the issues raised by the reviewers were addressed adequately. Therefore, the manuscript would be acceptable for publication.

Reviewer #4 (Remarks to the Author):

Reviewer #4 (Remarks to the Author):

The authors have a very interesting data set on the NH₃ uptake potential of PyOM. I would separate the conclusions into three parts:

- 1) The high uptake of NH₃ as a function of weathering and in contrast to NH₄
- 2) That this uptake involves covalent bonds
- 3) That these covalent bonds are heterocyclic molecules.

I think the degree of data support for the conclusions also tracks in this order. The first seems robustly supported. The second fairly well supported, and the third fairly thinly supported.

As previous reviewers have mentioned that only support for conclusion 3 is the NEXAFS spectra. While it is true as the authors rebut that this is a very good method to look at this, I think some more robust controls would be useful in order to support the conclusion. The beam damage test is helpful, but beam damage more quickly than detectable in this case is always a possibility. What would be very interesting would be some standards run on molecules analogous to those shown in the intermediate steps of figure 4, i.e. before aromatization of the N ring. In addition, some standards of classic Maillard products of NH₃ with non-aromatic carboxylic acid containing groups. It would be interesting both to get the spectra of these compounds and their susceptibility to beam damage. Given the potential resonant structures of some of the intermediate compounds shown in figure 4, it is not clear to me that they might not be resonant in the same general location of the functional groups assigned in figure 3. In addition, these compounds might be extremely susceptible to beam damage.

As other reviewers pointed out, it seems that the second step in figure 4, i.e. the conversion to aromatic N heterocycles, is a complex process, often necessitating the loss of O atoms, or C atoms, certainly intermediate disruption of the aromatic stabilization of the existing rings, and a whole series of complicated steps. The fact that these reactions would be spontaneous at room temperature is a fairly dramatic claim. On the other hand, an X-ray beam is just the sort of thing that could provide the necessary energy to drive this second step.

In addition, the authors need to provide a better explanation why these reactions would be spontaneous at 35°C on PyOM and not in other situations, i.e. why don't the intermediate molecules in Figure 4 always convert to the final molecules shown in figure 4.

I do think a reasonable explanation of most of the authors' data could stop with point 1 and point 2 above, and not necessitate point 3. I think this point needs more support.

Finally, as an editorial comment, the authors tend to slide between covalent bonds and nitrogen aromatic heterocycles in the manuscript somewhat loosely. I think that in the text the distinction needs to always be kept clear.

Thank you for your constructive comments on our manuscript. Please see our responses in red below.

Reviewer #1 (Remarks to the Author):

The authors have investigated pyrogenic organic matter (PyOM) reactions with ammonia showing ammonia retention under ambient conditions, and with ammonia retention altering with weathering. Further analysis demonstrates covalent bonding as a cause.

The manuscript is interesting and highly topical. I find the approach taken to be careful with appropriate statistical and quality checks on the study. As a consequence the conclusions are sound. The authors have responded to the original critiques of the reviewers carefully and rationally and demonstrated the originality of the current study.

I found no grammatical or typographic errors.

I think the authors have clearly presented some novel work. I recommend the study is published.

New response of the authors: Thank you for the kind words.

Reviewer #2 (Remarks to the Author):

Response of the authors: The processes described in these patents differ substantially from the process that we describe in our manuscript (see summary below). The authors of these patents use lignite instead of PyOM—two very different materials that have dissimilar chemical composition and geographical distribution. The authors also expose the lignite to NH₃ at temperatures reaching 100 C, which far exceeds ambient environmental temperatures. Furthermore, the authors do not show any evidence that covalent bonds form between NH₃ and lignite providing only anecdotal support, such as the claim that N forms contained in lignite “differ by their hydrolyzability.”

Summary: US Patent 6695892 states that amide bonds can form when lignite (also known as brown coal, made from compressed peat) is exposed to NH₃ at temperatures reaching 100 C. EP 1144342 states that amide, ammonium, “organically bonded,” and bonds “not hydrolysable as amide organically bonded” can form between lignite and NH₃ at temperatures reaching 100 C. EP 99968302.2 makes similar claims.

New comment of the reviewer:

Well, the difference between lignite and PyOM depends upon the source of PyOM. It is true that PyOM from wood produced during vegetation fires will contain a considerable amount of furan-type structures, whereas lignite contains more lignin derivatives. However, both contain aromatic structures

New response of the authors: We agree that the difference between lignite and PyOM depends upon the source of PyOM, as well as other conditions that the organic matter is exposed to during lignite or PyOM production. However, lignite is not a pyrogenic material (Hammes et al. 2007 GCB), and as noted in the comment above, will contain different chemical structures compared to PyOM. More importantly, the authors of the patents described above expose lignite to NH_3 at temperatures reaching 100C, which far exceeds ambient environmental temperatures. One of the key differences between our study and these patents is that we exposed PyOM to NH_3 at 35C.

Response of the authors: We were also surprised and excited to see that these reactions are possible at 35 C, a temperature that occurs naturally in ecosystems throughout the world. This comment supports our assertion of novelty.

New comment of the reviewer: If I am honest, it does not support your assertion of novelty but sincere doubts. What are the activation energies for forming covalent bonds? How was this activation energy reduced?

New response of the authors: We were also surprised to see that these reactions were possible at 35C. However, upon review of the literature, we found that other authors had observed the formation of non-cyclic amine and amide bonds when they exposed industrial relatives of PyOM to NH_3 at ambient temperatures even lower than 35C (Petit et al. 2009 JMC). Our work builds upon these published studies and shows that the formation of a wider variety of covalent bonds is also possible under ambient conditions, and that these bonds can form between NH_3 and PyOM materials that exist in the natural environment (and not only for industrial relatives of PyOM, as investigated for example in the cited study by Petit et al. 2009 JMC).

Response of the authors: We are not claiming that most soil N is bound in heterocyclic rings, but that NH_3 can interact with PyOM to form covalent bonds, including non-cyclic amides, heterocyclic rings, and other structures. Our experiments provide evidence that this is possible under laboratory conditions and indicate that it may also occur in the environment. We think that it is important to know this regardless of whether or not these N structures accumulate in soil, and whether or not most soil N is bound in amides. Additionally, multiple studies published over the past decade do show that cyclic N can accumulate in soils (Gillespie et al. 2014, Smernik & Baldock 2005). Although some of these cyclic N structures may be part of burnt residues, there are many other sources of cyclic N, including nucleic acids and amino acids, among others. However, because these points are beyond the goals and scope of our study, and we chose not to include them in our manuscript.

New Comments of the reviewer:

Other studies showed that heterocyclic N has low biochemical stability and there are a wide range of microorganisms which are able to degrade such compounds (i.e. Fetzner, Appl. Microbiol. Biotechnol. 49, 237-250, 1998). The contribution of Porphyrin from Chlorophyll is less than 1% of the dry mass of plants, the contribution of histidine to the total amino acid N in Grass is between 3 to 4% and that of DNA-N is not much higher.

Now, I was not saying that the authors are claiming that most soil N is bound in heterocyclic rings but if the proposed reaction occurs, there should be much more heterocyclic N in soils than presently identified. Thus the lack of higher amounts of heterocyclic N in unburnt soils points against the occurrence of the suggested reaction in natural soils.

New response of the authors: While we also think that the biochemical stability of heterocyclic N in soil is a very interesting topic, it is beyond the scope of our study. In our manuscript, we do not make any claims about the quantity of heterocyclic N that is found in soil, or about the ability of microorganisms to metabolize these compounds. We agree with the referee that these are important research questions and hope that additional studies will be conducted to investigate these topics (a suggestion that we have included at the end of our manuscript).

Response of the author: NEXAFS is an excellent way to investigate whether or not covalent bonding occurs. We considered using NMR in addition to NEXAFS, but since NMR does not accurately detect heterocyclic N in organic matter, we decided to employ the more recently developed NEXAFS spectroscopy, which utilizes the most advanced synchrotron system available (see Smernik & Baldock 2005; Leinweber et al. 2013).

New comment of the reviewer: It is simply not true that NMR does not accurately detect heterocyclic N in organic matter and there is no physical reason why it should be a problem, if correct acquisition parameters are applied. In the literature there are enough examples showing that NMR can be used for that purpose and several publications show solid-state ^{15}N NMR spectra of burnt soils with a dominance of heterocyclic N. However, even if one does not trust in the quantification of such spectra, the formation of such compounds by the proposed reaction should be indicated by an increasing signal in the respective chemical shift region, in particular if the proposed reactions is of such high importance as claimed (a general rule is that C and N should be visible by CPMAS NMR if the distance is not higher than two to three bonds.....)

New response of the authors: We chose to investigate the formation of covalent N bonds in our samples using NEXAFS spectroscopy because it is an excellent method for this purpose. We appreciate your interest in confirming the conclusions based on our NEXAFS data using another method. Following this suggestion, we collected new FTIR spectra from our samples (please see revised Fig. 3 below), which allowed us to generate information about a wide variety of functional groups present in PyOM, PyOM following exposure to NH_4^+ , and PyOM following exposure to NH_3 . We thank the referees for the suggestion to confirm our NEXAFS results by other spectroscopic techniques, a process which helped us to round out our interpretations and significantly enhanced the rigor of the conclusions. We do not think that it is necessary to add NMR spectra to the set of analyses presented in this manuscript, but appreciate the suggestion. The challenges associated with the use of NMR to generate information about cyclic N bonds does not reflect our own research, but is pervasive in the literature and may be shared by readers (e.g., Smernik & Baldock 2005 Biogeochemistry).

Reviewer #3 (Remarks to the Author):

The revised manuscript is greatly improved and the issues raised by the reviewers were addressed adequately. Therefore, the manuscript would be acceptable for publication.

New response of the authors: Thank you for your help in the revisions.

Reviewer #4 (Remarks to the Author):

Reviewer comment: The authors have a very interesting data set on the NH₃ uptake potential of PyOM. I would separate the conclusions into three parts:

- 1) The high uptake of NH₃ as a function of weathering and in contrast to NH₄
- 2) That this uptake involves covalent bonds
- 3) That these covalent bonds are heterocyclic molecules.

I think the degree of data support for the conclusions also tracks in this order. The first seems robustly supported. The second fairly well supported, and the third fairly thinly supported.

Reviewer comment: As previous reviewers have mentioned that only support for conclusion 3 is the NEXAFS spectra. While it is true as the authors rebut that this is a very good method to look at this, I think some more robust controls would be useful in order to support the conclusion. The beam damage test is helpful, but beam damage more quickly than detectable in this was always a possibility. What would be very interesting would be some standards run on molecules analogous to those shown in the intermediate steps of figure 4, i.e. before aromatization of the N ring. In addition, some standards of classic Maillard products of NH₃ with non-aromatic carboxylic acid containing groups. It would be interesting both to get the spectra of these compounds and their susceptibility to beam damage. Given the potential resonant structures of some of the intermediate compounds shown in figure 4, it is not clear to me that they might not resonate in the same general location of the functional groups assigned in figure 3. In addition, these compounds might be extremely susceptible to beam damage.

New response of the authors: Thank you for these insightful comments and suggestions.

New response of the authors: We made several major revisions to the text and included new data. We have revised the manuscript to include a more detailed description of our data collection conditions and beam damage tests and added the full series of spectra originally collected from each sample to assess whether beam damage occurred when samples were exposed to increasing radiation. We hope that adding this information will help clarify this issue. We have also conducted additional beam damage tests (summarized below).

New response of the authors: In order to further investigate the possibility of beam damage, we conducted another round of beam damage tests on our oxidized PyOM sample that had been exposed to NH₃. Although there does not appear to be any evidence of beam damage in the spectra that we present in the manuscript (see revised version of Fig. 3 and Supplementary Fig. 8—formerly Supplementary Fig. 7, both copied below), we did find that beam damage is possible when we expose the sample to much higher radiation (see Figs A and B below). However, this requires a substantial increase in radiation compared to what we used originally.

To collect the spectra presented in Fig. 3, we used 20-second slew scans with the exit slit set to 15 μm . Each scan was collected from a new location on the sample, so that exposure to radiation was kept to the lowest possible dose. For the beam damage tests now presented in Supplementary Fig. 8, we collected a series of 15 scans from the same location, in order to check whether repeated exposure to radiation introduced artifacts into the spectra. If the sample was susceptible to beam damage, we would expect to see new spectral features that would become more pronounced as the sample was exposed to more radiation. However, as shown in Supplementary Fig. 8, this did not occur—even after a 15-fold increase in radiation, no new spectral features developed. For the new beam damage tests, we used 60-second slew scans with the exit slit set to 50 μm , which represents a dose of radiation that is an order or magnitude higher than before. As shown in Figs A and B (below), there are two peaks that appear to increase as the sample is exposed to greater radiation.

Fig. 3. Nitrogen K -edge NEXAFS and FTIR spectra of oxidized PyOM samples. (a) NEXAFS spectra of oxidized PyOM, oxidized PyOM following exposure to NH_4^+ , and oxidized PyOM following exposure to NH_3 (see Supplementary Fig. 4 for NEXAFS spectra of all PyOM samples scaled by N content). Colored bars represent the range of peak centers associated with selected spectral features: 397.75-398.8 for C=N bonds in 1N and 2N aromatic six-membered rings (red), 400 for nitrile bonds (orange), 399.76-400.27 for C=N bonds in 2N five-membered rings (yellow), 401.2-402.4 for C-N bonds in 1N and 2N aromatic five-membered rings (green), 403-

403.75 for N bonded to aromatic rings (blue), and 405-406.58 for N-H bonds (purple). Model chemical structures are shown at the top of the figure. (b) FTIR spectra of oxidized PyOM, oxidized PyOM following exposure to NH_4^+ , and oxidized PyOM following exposure to NH_3 .

Supplementary Figure 8. Beam damage tests for N K-edge NEXAFS spectra. Fifteen scans were collected from the same location in each sample (PyOM, PyOM following exposure to NH_4^+ , and PyOM following exposure to NH_3), thereby exposing each sample to fifteen-fold the dose of radiation that was used for other analyses (e.g., spectra presented in Fig. 3a and deconvolution results presented in Supplemental Table 2). Spectra above are arranged in the order that they were collected, so that the first scan collected is shown at the bottom of the figure and the following fifteen scans are shown in ascending order. An average of all fifteen scans is shown in the panel at the top of the figure. Repeated exposure to X-ray radiation did not create notable changes in spectral features, indicating that beam damage did not occur during collection of NEXAFS spectra. The averaged spectra in the top panel do not contain spectral features that are distinct from the averaged spectra collected from different locations in each sample (Fig. 3a), further indicating that exposure to X-rays did not damage the samples or introduce artifacts into the NEXAFS spectra collected from these samples.

Figure A (above left). Nitrogen K -edge NEXAFS spectrum of an oxidized PyOM sample following exposure to NH_3 . The spectrum represents an average of 70 scans that were each collected from a new location of the sample.

Figure B (above center). Series of nitrogen K -edge NEXAFS spectra collected from the same location of an oxidized PyOM sample following exposure to NH_3 . A total of 120 scans were collected from the same location. The four spectra each show the average of ~ 30 of these scans, grouped by progressive exposure to radiation and displayed in ascending order: (a) scans 1-29, (b) scans 30-59, (c) scans 60-89, (d) scans 90-120.

The spectrum from glycine anhydride (Supplementary Fig. 6, also copied below), a compound that is very similar to the predicted intermediates shown in Fig. 4, helps us address the concern expressed above, as well. Under the conditions that we used to collect the NEXAFS spectra presented in our manuscript (i.e., 20-second slew scans collected from fresh locations on each sample or standard compound, and a relatively narrow exit slit setting), we do not see any signs of beam damage in the glycine anhydride spectrum, nor do glycine anhydride's N bonds seem to resonate in the same locations associated with aromatic 6-membered heterocycles. This suggests that similar compounds are not susceptible to beam damage under the data collection conditions that we used, and that their presence can be distinguished from the presence of aromatic heterocycles in our PyOM samples.

Supplementary Figure 6. N K-edge NEXAFS spectra of N standard compounds. See Supplementary Table 3 for specific peak values.

Reviewer comment: As other reviewers pointed out, it seems that the second step in figure 4, i.e. the conversion to aromatic N heterocycles, is a complex process, often necessitating the loss of O atoms, or C atoms, certainly intermediate disruption of the aromatic stabilization of the existing rings, and a whole series of complicated steps. The fact that these reactions would be spontaneous at room temperature is a fairly dramatic claim. On the other hand, and X-ray beam is just the sort of thing that could provide the necessary energy to drive this second step.

New response of the authors: Based on our analyses, there is no evidence that the level of radiation that our samples were exposed to in this study was sufficient to drive the aromatization of non-aromatic heterocyclic structures. In order to further investigate the possibility of beam damage from NEXAFS spectroscopy, we also collected FTIR spectra from the same set of samples that was originally used for Fig. 3 (see revised Fig. 3 above) and compared these spectra to the FTIR spectra of NH_3 and with NH_4^+ (which contain predominant peaks around 950 and 1440 cm^{-1} ; see NIST WebBook cited in manuscript). These new FTIR data support the conclusions based on our previous NEXAFS data—that exposure to NH_3 results in the formation of a variety of covalent bonds between NH_3 -N and PyOM that could not be accounted for solely through NH_3 adsorption or electrostatic interactions with NH_4^+ . Similar to the NEXAFS spectra, the FTIR spectra show clear differences between bonds present in PyOM, PyOM following exposure to NH_3 , and PyOM following exposure to NH_4^+ . Exposure to NH_3 resulted in the emergence of new peaks at 1656 and 1367 cm^{-1} and an increase in peak height at 1574 cm^{-1} , all of which may be associated with C-N stretching, including C-N stretching in aromatic rings at 1656 cm^{-1} . Exposure to NH_3 also resulted in a marked decrease in the peaks associated with C=O and C-O carbonyl/carboxyl and ketonic stretching at 1710 and 1226 cm^{-1} , respectively, suggesting that these functional groups decrease relative to N-containing functional groups in this sample. In contrast, exposure to NH_4^+ resulted in only two new spectral features, including the formation of medium-sized peaks at 1419 and 1372 cm^{-1} , which may be associated with N-H and C-N stretching, respectively. Since FTIR spectroscopy subjects samples to much lower energy than NEXAFS, the risk of beam damage is even lower. Taken together, these data support our original conclusion that a variety of covalent bonds, including heterocyclic aromatic structures, can form when oxidized PyOM is exposed to NH_3 under ambient conditions.

Reviewer comment: In addition, the authors need to provide a better explanation why these reactions would be spontaneous at 35C on PYOM and not in other situations, i.e. why don't the intermediation molecules in Figure 4 always convert to the final molecules shown in figure 4.

New response of the authors: We did not intend for the reactions presented in Fig. 4 to be interpreted as results from our experiments and agree with the referee that this should be explained in the manuscript. Our discussion of interactions between NH_3 and PyOM in the environment benefit from such a figure, but our experiments only provide evidence that the final products can form—not the exact pathway that resulted in the formation of these products. This figure was meant to show the pathways that other authors have proposed to occur between NH_3 and industrially-produced relatives of PyOM. We have revised the language

in the manuscript to better reflect this. We are happy to make additional changes to address this issue if the current manuscript is not sufficiently clear (including relegating it to the supplementary material).

Reviewer comment: I do think a reasonable explanation of most of the authors data could stop with point 1 and point 2 above, and not necessitate point 3. I think this point needs more support.

New response of the authors: Thank you very much for making a constructive suggestion for the way forward. We hope by (i) including new FTIR data, (ii) revising the display of the beam damage tests and its description, (iii) conducting additional beam damage tests, and (iv) revising the manuscript to more clearly state that Fig. 4 data are not our proposed mechanisms but hypothesized by others (all four changes are described in more detail in the above replies and supported by figures), the storyline becomes clearer and defensible. We are open to discussions here we have still deficiency in our chain of arguments.

Reviewer comment: Final as an editorial comment, the authors tend to slide between covalent bonds and nitrogen aromatic heterocycles in the manuscript somewhat loosely. I think that in the text the distinction needs to always be kept clear.

New response of the authors: Thank you for your careful reading of the manuscript. We agree that these terms are not fully interchangeable and have revised the manuscript in order to try to ensure appropriate usage. Our understanding of the terminology is outlined below (and we revised the manuscript to be consistent in its use), but we are open to making changes.

Covalent: a bond involving atoms that share electron pairs; used in the manuscript as a general term, including single, double, and triple bonds, regardless of whether atoms are present in aliphatic, cyclic, or aromatic compounds

Heterocycle, heterocyclic compound: a cyclic structure containing at least two elements (C and N, in our case); can include aromatic or non-aromatic ring structures

Aromatic: a cyclic structure with a ring of resonance bonds

Reviewers' comments:

Reviewer #2 (Remarks to the Author):

I greatly appreciate the effort the authors conducted to produce additional evidence for their statement there astonishing observation that PyOM and Ammonia reacts at ambient conditions to form N-heteroaromatic structures.

However, the new evidences by FT-IR still do not convince me, since FT-IR is known to for considerable overlapping of IR bands. I am not an expert in FT-IR and thus after a literature search, I consulted several of my colleagues, which are experts and they confirmed that the band at 1656 cm⁻¹ is very typical for C=O in amides but can also contain contributions from aromatic structures. Thus, it is not an unbiased proof for heterocyclic aromatic n. Ammonium ions result in bands around 1423 cm⁻¹ and further contributions of aromatic compounds may occur around 1590 cm⁻¹. Further, consulting my old chemistry books, I could read that ethanoic acid reacts with excess ammonia to give ammonium ethanoate. After heating the ammonium salt dehydrates producing ethanamide. Thus, already the step of amide formation needs some activation energy. But - well - if amides are formed they would also contribute to the signal at 1656 cm⁻¹.

Only recently, we did a study on charcoal from wood, which was incubated with urea and ammonia, although I am aware that there is a big and important difference because our biochars were not oxidized before the incubation. Applying 15N NMR spectroscopy we could not detect the formation of heterocyclic N during the latter, although we were able to confirm the formation of heterocyclic aromatic N during the charring of N-rich feedstocks, such as sewage sludge, casein or chitin or grass material. We applied Bloch Decay techniques, which do not depend upon proton content to the charred grass and were able to confirm that routine solid-state can indeed observe non-protonated N. The latter confirms that heterocyclic aromatic N can be detected by solid-state 15N NMR spectroscopy. At least our group is able to do it, and we do this since about 25 years; Actually, we are not the only one who can. I am pretty well informed about the literature of NMR on coal (1) and burnt and unburnt pyrogenic matter in soil and I can tell you that it is not pervasively stated in the literature that solid-state 15N NMR spectroscopy cannot detect heterocyclic aromatic N. Thus, the claim that 15N NMR was not performed because it doesn't result in valuable data is a bit weak argumentation.

However, I know that solid-state 15N NMR spectroscopy is not an easy technique and the authors may have to use 15N enriched Ammonia to obtain spectra of sufficient quality, but it would be enable to distinguish between pyrrole-type N (chemical shift relative to nitromethane scale: -150 to -240 ppm) from amides (-240 to 285 ppm), pyridine N (around -60 to -80 ppm), protonated N in pyridine-type structures (shift toward: -150 to -240 ppm, but will only be present at alkaline conditions)(2). Ammonia will not be visible, but ammonium could be detected at -358 ppm. Thus, the respective signals can be separated and distinguished and the proof would be unbiased. Maybe it is the only technique which allows this and if organic structures are formed, the right 15N NMR parameters are used and the N concentration is higher than 1% (in case of natural abundance) then signals should appear after 1 million scans.

2)Witanowski, M., Stefaniak, L., Webb., G.A., 1993. Nitrogen NMR Spectroscopy. In: Annual Reports on NMR Spectroscopy, 25 (Ed. by G. Webb), pp. 480. Academic Press, London.

1)Knicker, H., Hatcher, P.G., Scaroni, A.W., 1996. A solid-state 15N NMR spectroscopic investigation of the origin of nitrogen structures in coal. International Journal of Coal Geology, 32, 255-278

Reviewer #4 (Remarks to the Author):

I appreciate the efforts the authors have made to address the reviewers comments. I think there is much of interested in the manuscript.

Unfortunately, I still do not think that the quality of evidence for aromatic N compounds forming rise to the level of the claim.

The authors offer glycine anhydride as an example of why non-aromatic structures couldn't be confused for aromatic structures in the nexafs data. This seems to have two problems: 1) while the authors are careful to say that glycine anhydride doesn't resonate where 6 membered aromatic rings are assigned in their work, the large resonance of glycine anhydride does appear to be exactly in the range assigned by the authors for 5 membered aromatic rings. 2) glycine anhydride is not a particularly conjugated amide. What about a compound like benzamide? That is more analogous to the intermediates shown in the paper and the sort of thing you might expect to see in PyOM. Again, thinking of different types of amides, the ir assignment given by the authors of 1656 is right in the range of C=O stretches in amides that run from ~1650-1690. I really think the authors need to exclude other conjugated N containing molecules in order to claim N aromatics.

Reviewer #2 (Remarks to the Author):

I greatly appreciate the effort the authors conducted to produce additional evidence for their statement there astonishing observation that PyOM and Ammonia reacts at ambient conditions to form N-heteroaromatic structures.

However, the new evidences by FT-IR still do not convince me, since FT-IR is known to for considerable overlapping of IR bands. I am not an expert in FT-IR and thus after a literature search, I consulted several of my colleagues, which are experts and they confirmed that the band at 1656 cm^{-1} is very typical for C=O in amides but can also contain contributions from aromatic structures. Thus, it is not an unbiased proof for heterocyclic aromatic n. Ammonium ions result in bands around 1423 cm^{-1} and further contributions of aromatic compounds may occur around 1590 cm^{-1} . Further, consulting my old chemistry books, I could read that ethanoic acid reacts with excess ammonia to give ammonium ethanoate. After heating the ammonium salt dehydrates producing ethanamide. Thus, already the step of amide formation needs some activation energy. But - well - if amides are formed they would also contribute to the signal at 1656 cm^{-1} .

Response: Thank you for your comments. We agree that functional group assignment using FTIR is challenging because the regions associated with different bonds often overlap with one another. It is for this reason that we originally used NEXAFS spectroscopy and more recently used NMR spectroscopy to investigate the functional group composition of our samples. In contrast to FTIR, both of these methods are element-specific and offer higher spectral resolution. We appreciate your suggestion to use NMR and have included more detailed information below and in the manuscript.

With respect to the FTIR spectra: we agree that in addition to being typically associated with amidic C=O bonds, the peak at 1656 cm^{-1} may be associated with aromatic structures. However, the only treatment difference between the PyOM samples presented in Fig. 3 is whether or not they were exposed to NH_3 or NH_4^+ . Therefore, it is likely that the emergence of an FTIR peak at 1656 cm^{-1} in the PyOM+ NH_3 spectrum—a peak which is not present in the spectra collected from unexposed PyOM or PyOM+ NH_4^+ —is a result of interactions between PyOM and NH_3 -N, rather than a signal from the aromatic C structures which are present in all three samples and cannot be generated by exposure to NH_3 . We agree that the peaks around 1419-1423 cm^{-1} may be associated with N-H bonds, either from NH_4^+ or other functional groups containing N-H bonds, and have noted in the manuscript that exposure to NH_4^+ results in increased peak height in this region. We also agree that the peak at 1590 cm^{-1} can be associated with aromatic compounds. There is a strong signal around 1590 cm^{-1} in the FTIR spectra collected from our oxidized PyOM and PyOM+ NH_4^+ samples, suggesting that the PyOM samples contain aromatic structures and that exposure to NH_4^+ does not have a strong effect on the relative proportion of these structures. However, in the spectrum collected from the PyOM+ NH_3 sample, the peak at 1590 cm^{-1} is overtaken by one at 1574 cm^{-1} , which may be associated with C-N bonds.

As discussed above, we agree that FTIR alone should not be used to determine whether covalent bonds form between PyOM and NH_3 -N. However, combined with NEXAFS, NMR, and

elemental analyses, the FTIR spectra support our conclusions that PyOM exposure to NH_3 results in changes to PyOM functional group composition that do not occur when PyOM is exposed to NH_4^+ .

Only recently, we did a study on charcoal from wood, which was incubated with urea and ammonia, although I am aware that there is a big and important difference because our biochars were not oxidized before the incubation. Applying ^{15}N NMR spectroscopy we could not detect the formation of heterocyclic N during the latter, although we were able to confirm the formation of heterocyclic aromatic N during the charring of N-rich feedstocks, such as sewage sludge, casein or chitin or grass material. We applied Bloch Decay techniques, which do not depend upon proton content to the charred grass and were able to confirm that routine solid-state can indeed observe non-protonated N. The latter confirms that heterocyclic aromatic N can be detected by solid-state ^{15}N NMR spectroscopy. At least our group is able to do it, and we do this since about 25 years; Actually, we are not the only one who can. I am pretty well informed about the literature of NMR on coal (1) and burnt and unburnt pyrogenic matter in soil and I can tell you that it is not pervasively stated in the literature that solid-state ^{15}N NMR spectroscopy cannot detect heterocyclic aromatic N. Thus, the claim that ^{15}N NMR was not performed because it doesn't result in valuable data is a bit weak argumentation.

However, I know that solid-state ^{15}N NMR spectroscopy is not an easy technique and the authors may have to use ^{15}N enriched Ammonia to obtain spectra of sufficient quality, but it would be enable to distinguish between pyrrole-type N (chemical shift relative to nitromethane scale: - 150 to -240 ppm) from amides (-240 to 285 ppm), pyridine N (around -60 to -80 ppm), protonated N in pyridine-type structures (shift toward: -150 to -240 ppm, but will only be present at alkaline conditions)(2). Ammonia will not be visible, but ammonium could be detected at -358 ppm. Thus, the respective signals can be separated and distinguished and the proof would be unbiased. Maybe it is the only technique which allows this and if organic structures are formed, the right ^{15}N NMR parameters are used and the N concentration is higher than 1% (in case of natural abundance) then signals should appear after 1 million scans.

Response: Thank you for this suggestion. As you recommended, we prepared oxidized PyOM samples for ^{15}N -NMR spectroscopy by exposing them to either $^{15}\text{N}\text{-NH}_4^+$ or $^{15}\text{N}\text{-NH}_3$ (see Fig. 3c). Since we used NH_3 as the reference compound rather than nitromethane, we can convert the peak assignment values listed above using nitromethane as a reference compound by adding +380 ppm. Consistent with our NEXAFS and FTIR spectra and elemental analyses, we see evidence that oxidized PyOM retains substantial quantities of $^{15}\text{NH}_3\text{-N}$ via several mechanisms. The ^{15}N -NMR spectra suggest that the dominant $\text{NH}_3\text{-N}$ retention mechanisms may include NH_4^+ adsorption (represented by the peak at ~ 20 ppm), and the formation of covalent bonds such as amines (~ 20 ppm), amides (~ 107 ppm), and heterocyclic pyrroles and indoles (overlapping peaks at $\sim 130\text{-}165$ ppm). Based on NMR peak area integration, it appears that following exposure to $^{15}\text{NH}_3$, more than 40% of the N is retained through the formation of covalent C-N bonds, 13% of which may be heterocyclic structures. This is similar to our NEXAFS

deconvolution results, which suggest that ~25% of NH₃-N is retained in N heterocycles. Also consistent with our previous analyses, the ¹⁵N-NMR spectrum collected from PyOM+NH₄⁺ does not show significant signs of covalent bond formation between PyOM and ¹⁵NH₄⁺-N. It is possible that these samples do contain covalently bound N, but that they play no major role.

We have also conducted stoichiometric analyses, which further support both the NEXAFS and NMR data. Together, these analyses suggest that much of the N retained by PyOM following exposure to NH₃ is incorporated into heterocyclic rings (see pages 6 and 7 of the commented manuscript).

Thank you again for your comments. We think that your input and the addition of NMR spectroscopy have greatly strengthened the manuscript.

- 2)Witanowski, M., Stefaniak, L., Webb., G.A., 1993. Nitrogen NMR Spectroscopy. In: Annual Reports on NMR Spectroscopy, 25 (Ed. by G. Webb), pp. 480. Academic Press, London.
- 1)Knicker, H., Hatcher, P.G., Scaroni, A.W., 1996. A solid-state ¹⁵N NMR spectroscopic investigation of the origin of nitrogen structures in coal. International Journal of Coal Geology, 32, 255-278

Reviewer #4 (Remarks to the Author):

I appreciate the efforts the authors have made to address the reviewers comments. I think there is much of interested in the manuscript.

Unfortunately, I still do not think that the quality of evidence for aromatic N compounds forming rise to the level of the claim.

The authors offer glycine anhydride as an example of why non-aromatic structures couldn't be confused for aromatic structures in the nexafs data. This seems to have two problems: 1) while the authors are careful to say that glycine anhydride doesn't resonate where 6 membered aromatic rings are assigned in their work, the large resonance of glycine anhydride does appear to be exactly in the range assigned by the authors for 5 membered aromatic rings. 2) glycine anhydride is not a particularly conjugated amide. What about a compound like benzamide? That is more analogous to the intermediates shown in the paper and the sort of thing you might expect to see in PyOM. Again, thinking of different types of amides, the ir assignment given by the authors of 1656 is right in the range of C=O stretches in amides that run from ~1650-1690. I really think the authors need to exclude other conjugated N containing molecules in order to claim N aromatics.

Response: Thank you for your comments. We agree that there is some overlap between NEXAFS spectral features, including one of the major NEXAFS features associated C-N bonds in nonaromatic 6-membered N heterocycles such as glycine anhydride (401.15 eV) and with C-N bonds in some aromatic 5-membered heterocycles (401.43 eV). However, there are several additional features associated with 5- and 6-membered aromatic N heterocycles present in our

PyOM+NH₃ NEXAFS spectra, including several peaks below 400 eV. Aromatic 5-membered heterocycles resonate around 400.05 eV (for C=N bonds) and at 402.4 eV (for C-N in 1N heterocycles), which are far enough away to be distinguished from the peak associated with C-N bonds in nonaromatic 6-membered heterocycles at 401.15 eV. To our knowledge, the only N functional groups that resonate between 397-399 eV are aromatic heterocycles. Compounds that contain benzene rings and aliphatic nitrogen (such as 4-nitrophenylacetic acid and benzamide) resonate at 400 eV and higher^{1,2} and cannot be responsible for the distinct features below 400 eV present in the NEXAFS spectra collected from PyOM following exposure to NH₃.

In order to further investigate the presence of aromatic N heterocycles, we also collected ¹⁵N-NMR spectra from our samples (see Fig. 3c). These spectra suggest that aromatic N heterocycles such as pyrroles and indoles form when PyOM is exposed to NH₃ (overlapping peaks at ~130-165 ppm). Based on NMR peak area integration, it appears that following exposure to ¹⁵NH₃, approximately 13% of the N is retained through the formation of aromatic N heterocycles. This is similar to our NEXAFS deconvolution results, which suggest that ~25% of NH₃-N is retained in N heterocycles (including both aromatic and non-aromatic heterocycles). Also consistent with our previous analyses, the ¹⁵N-NMR spectrum collected from PyOM+NH₄⁺ does not show significant signs of covalent bond formation between PyOM and ¹⁵NH₄⁺-N. It is possible that these samples do contain covalently bound N, but they do not play any quantitatively important role.

We have also conducted stoichiometric analyses, which further support both the NEXAFS and NMR data. Together, these analyses suggest that much of the N retained by PyOM following exposure to NH₃ is incorporated into heterocyclic rings (see pages 6 and 7 of the commented manuscript).

With respect to the FTIR spectra: we agree that the peak at 1656 cm⁻¹ may be associated with aromatic structures. However, the only treatment difference between the PyOM samples presented in Fig. 3 is whether or not they were exposed to NH₃ or NH₄⁺. Therefore, it is likely that the emergence of an FTIR peak at 1656 cm⁻¹ in the PyOM+NH₃ spectrum—a peak which is not present in the spectra collected from unexposed PyOM or PyOM+NH₄⁺—is a result of interactions between PyOM and NH₃-N, rather than a signal from the aromatic structures which are present in all three samples and cannot form by exposure to NH₃. Regardless, we agree that FTIR alone should not be used to determine whether covalent bonds form between PyOM and NH₃-N. However, combined with NEXAFS, NMR, and elemental analyses, the FTIR spectra support our conclusions that PyOM exposure to NH₃ results in changes to PyOM functional group composition that do not occur when PyOM is exposed to NH₄⁺, and that aromatic N heterocycles are among the structures formed.

Thanks again for your comments—your input has greatly strengthened the manuscript.

References

1. Leinweber, P. *et al.* Nitrogen K-edge XANES - An overview of reference compounds used to identify 'unknown' organic nitrogen in environmental samples. *J. Synchrotron Radiat.* **14**, 500-511 (2007).
2. Stevens, J.S., Gainar, A., Jaye, C., Fischer, D.A., and Schroeder, S.L.M. NEXAFS and XPS of p-Aminobenzoic Acid Polymorphs: The Influence of Local Environment. *J. Phys.: Conf. Ser.* **712** 012133 (2016).

Reviewers' comments:

Reviewer #1 (Remarks to the Author):

This is a well written, clear concise, manuscript that increases our knowledge surrounding the chemistry that occurs when PyOM is exposed to NH₃.

My review is focused on how the authors responded to reviewers 2 and 4.

Reviewers 2 and 4 have continued to raise concerns regarding the strength of the evidence supporting the presence of heterocyclic N aromatics.

I think the authors have presented additional and strong irrefutable evidence to support their earlier conclusions. They have provided this evidence by doing further work as reviewer 2 suggested: they have applied a ¹⁵NMR technique to demonstrate PyOM retains ¹⁵NH₃-N.

Reviewers 2 and 4 questioned if the earlier manuscript's evidence supported heterocyclic aromatic N. The authors FTIR data provided strong evidence but not conclusive evidence for this. Following the additional experimentation with ¹⁵NH₃ there is clear evidence or "unbiased proof" that NH₃-N has moved into heterocyclic compounds.

The reviewers comments AND the uptake of the ¹⁵N method by the authors has greatly strengthened the paper.

I believe this is ready to publish.

Reviewer #5 (Remarks to the Author):

After careful reading of the manuscript, I have some serious doubts about the data. First of all, based on my experience, the CPMAS ¹³C NMR spectra present some problems which I would like to have clarified by the authors. In addition, it is not clear to me which kind of mechanism is behind the formation of the covalent bonds the authors describe in the paper. Please find my comments in the attached file. I think that the paper is not at the moment eligible for publication in its present form.

Submission of revised Manuscript NCOMMS-16-03375

Reviewer #1 (Remarks to the Author):

This is a well written, clear concise, manuscript that increases our knowledge surrounding the chemistry that occurs when PyOM is exposed to NH₃.

My review is focused on how the authors responded to reviewers 2 and 4.

Reviewers 2 and 4 have continued to raise concerns regarding the strength of the evidence supporting the presence of heterocyclic N aromatics.

I think the authors have presented additional and strong irrefutable evidence to support their earlier conclusions. They have provided this evidence by doing further work as reviewer 2 suggested: they have applied a 15NMR technique to demonstrate PyOM retains 15NH₃-N.

Reviewers 2 and 4 questioned if the earlier manuscript's evidence supported heterocyclic aromatic N. The authors FTIR data provided strong evidence but not conclusive evidence for this. Following the additional experimentation with 15NH₃ there is clear evidence or "unbiased proof" that NH₃-N has moved into heterocyclic compounds.

The reviewers comments AND the uptake of the 15N method by the authors has greatly strengthened the paper.

I believe this is ready to publish.

Response: Thank you for supporting our work.

Reviewer #5 (Remarks to the Author):

Here there is something unclear to me. The authors state that, following oxidation, PyOM pH decreased. This is trivial because the oxygen containing functions may have acid properties. That means that an acid-base reaction between NH₃ and PyOM surface has been supposed by the authors. This reaction should be the responsible for an electrostatic interaction between ammonium and PyOM. My point is: I do not understand why the addition of HCl should increase ammonia retention. In fact, I can imagine that the addition of a strong acid should protonate the surface thereby leading to the formation of positive charges (e.g. -OH₂⁽⁺⁾). In the mean time NH₃ should be transformed in NH₄⁺. And then? Why do the author expect a change in the adsorption properties? Probably the authors want to state that the addition of HCl should decrease N adsorption, while they do not observe any change. Maybe they should explain better the point.

Response: Thank you for drawing our attention to this issue, which may have been unclear in the manuscript. This point was meant to address the incomplete understanding of PyOM-NH₃ interactions currently presented in the literature, which suggests that pH is the primary characteristic responsible for PyOM's NH₃ retention capacity and that the main mechanism for retention is through formation of NH₄⁺ salts and subsequent electrostatic interactions between NH₄⁺ and PyOM. In our manuscript, we make significant contributions to this topic by demonstrating that it is PyOM's oxygen-containing functional

groups (whose presence is often, but not always, positively correlated with PyOM acidity) rather than PyOM pH that contribute to its NH₃ retention capacity. Furthermore, we demonstrate that in addition to retention through electrostatic interactions, NH₃-N is also retained through covalent C-N bond formation. We have amended the text to clarify this point.

For what concerns the quantitative assessment of NMR spectra, please read my comments in the Material section

Response: Please see reply below.

It is not clear to me why exposition of biochar to NH₃ would lead to formation of covalent bonds. Although the authors are providing proofs that this could happen, I cannot understand which can be the mechanism by which ammonium could form covalent bonds in environmental conditions. By reading sample preparation, I still have not clear which can be the possible mechanism driving formation of covalent bounds between NH₃ and PyOM

Response: This is indeed, on the surface, a surprising outcome that has not been considered by most scientists investigating PyOM-N interactions and may direct us to rethink PyOM material science and N biogeochemistry on local and global scales. It should be noted that the reaction of amines (including ammonia as the simplest case) with carboxylic acids to form amides is very well known—it is in fact the basis of protein synthesis from amino acids. It is also well established that in some situations further reactions, including condensation and cyclization and aromatization to form N-aromatics, are also possible. The surprising thing, at face value, is that this might also occur rapidly at ambient environmental temperature under seemingly benign conditions. However, the Paal-Knorr pyrrole synthesis reaction—which produces pyrroles through the condensation of a dicarbonyl compound with an amine or ammonia—is thought to be responsible for N heterocycle formation between secondary organic aerosols and ammonia or amines in the atmosphere. Furthermore, although we have not investigated the specific chemical reactions ourselves yet, we have searched for mechanisms in the literature. See below for some published papers that include diagrams of proposed reactions between NH₃ and organic compounds.

Jansen, R. J. J. & Van Bakkum, H. Amination and ammoxidation of activated carbons. *Carbon* 32, 1507-1516 (1994).

Seredych, M. & Bandosz, T. J. Mechanism of ammonia retention on graphite oxides: Role of surface chemistry and structure. *J. Phys. Chem. C* 111, 15596-15604 (2007).

Hulicova-Jurcakova, D. et al. Nitrogen-enriched nonporous carbon electrodes with extraordinary supercapacitance. *Adv. Funct. Mater.* 19, 1800-1809 (2009).

Shen, W. & Fan, W. Nitrogen-containing porous carbons: synthesis and application. *J. Mater. Chem. A* 1, 999-1013 (2013).

Updyke, K. M., Nguyen, T. B., & Nizkorodov, S. A. Formation of brown carbon via reactions of ammonia with secondary organic aerosols from biogenic and anthropogenic precursors. *Atmos. Environ.* 63, 22-31 (2012).

De Haan, D. O. et al. Secondary Organic Aerosol-Forming Reactions of Glyoxal with Amino Acids. *Environ. Sci. Technol.* 43, 2818-2824 (2009).

Amarnath, V., Anthony, D. C., Amarnath, K., Valentine, W. M., Wetterau, L. A., & Graham, D. G. Intermediates in the Paal-Knorr synthesis of pyrroles. *J. Org. Chem.* 56, 6924-6931 (1991).

I have some problems in considering the NMR experiments as quantitative. First of all, all the conditions that the authors used do not allow any quantitative evaluation. Secondly, how did they account for the spinning side bands? I find strange that no SSB are revealed in the spectra, if one considers that pyrogenic carbon is mainly aromatic. Then, in my experience, 1 ms as contact time is too short for this type of organic systems. Also the recycle delays of only 3 s (for CPMAS) and 10 s (for SEDP) are too short. How did the authors chose those parameters? I think tha the only way to get quantitative information is by running direct polarization pulse sequence. However, I understand that this kind of experiment is too time consuming, but the experiments on which the authors base their discussion and conclusions are based on the assumption that all the samples reveal the same relaxing dynamics. This can be or cannot be the case. How can the authors prove this assumption?

Response: The reviewer is justified in raising the issue of quantitation in NMR analysis of such materials. The authors are acutely aware of the limitations and potential pitfalls, and have been at the forefront of both raising these concerns and developing ways to deal with these issues in a series of highly-cited papers on various aspects of quantitation in solid-state NMR, including ^{13}C and ^{15}N NMR analysis of organic matter in general:

Smernik, R.J. and Oades, J.M. (2000a). The use of spin counting for determining quantitation in ^{13}C NMR spectra of natural organic matter. 1. Model systems and the effects of paramagnetic impurities. *Geoderma*, 96, 101-129.

Smernik, R.J. and Oades, J.M. (2000b). The use of spin counting for determining quantitation in solid state ^{13}C NMR spectra of natural organic matter. 2. HF treated soil fractions. *Geoderma*, 96, 159-171.

And the specific issues relating to charred materials:

Smernik, R.J., Baldock, J.A., Oades, J.M. and Whittaker, A.K. (2002a). Determination of $T_{1\rho}\text{H}$ relaxation rates in charred and uncharred wood and consequences for NMR quantitation. *Solid State Nuclear Magnetic Resonance*, 22, 50-70.

Smernik, R.J., Baldock, J.A. and Oades, J.M. (2002b). Impact of remote protonation on ^{13}C CPMAS NMR quantitation of charred and uncharred wood. *Solid State Nuclear Magnetic Resonance*, 22, 71-82.

And the specific problems relating to solid-state ^{15}N NMR:

Smernik, R.J. and Baldock, J.A. (2005a). Does solid-state ^{15}N NMR spectroscopy detect all soil organic nitrogen? *Biogeochemistry*, 75, 507-528.

Smernik, R.J. and Baldock, J.A. (2005b). Solid-state ^{15}N NMR analysis of highly ^{15}N -enriched plant materials. *Plant and Soil*, 275, 271-283.

In order to directly address the questions raised regarding spinning side bands, contact time, recycle delays, relaxing dynamics, and quantitation, we conducted several new ^{13}C and ^{15}N NMR experiments and expanded the data and discussion in the revised manuscript (see Figs A-F below; all 6 new figures were also added to the Supplementary Information, as readers may have similar questions about the NMR spectra). The results of these new NMR experiments are described in greater detail below. Thanks to the reviewer for raising these questions and prompting a more thorough examination of the parameters used during data collection. We feel that this addition has greatly strengthened the manuscript.

First of all, all the conditions that the authors used do not allow any quantitative evaluation.

Response: Using ^{13}C NMR analysis, we confirmed the trends observed with FTIR and quantitative elemental analyses: oxidation resulted in an increase in the relative proportion of oxygen-containing functional groups present. Using ^{15}N NMR analysis, we confirmed the trends observed with FTIR, NEXAFS, and quantitative elemental analyses: exposure to NH_3 resulted in the formation of a variety of covalent bonds, including heterocyclic structures. Please see the text and Figs A-F below, which have been added to show that the conditions that we used to collect NMR spectra were appropriate for relative quantitation of the functional groups present in our samples.

Secondly, how did they account for the spinning side bands? I find strange that no SSB are revealed in the spectra, if one considers that pyrogenic carbon is mainly aromatic.

Response: Thank you for bringing this up. Spinning side-bands (SSB) occur when the rate of magic angle spinning (MAS) is less than the chemical shift anisotropy (CSA). Under non-spinning conditions, solid-state NMR spectra of nuclei in unsymmetrical (anisotropic) chemical environments are broad and often unsymmetrical. ^{13}C nuclei in aromatic structures are a classic example, as shielding of the nuclei provided by the π -electrons varies strongly with the orientation of the aromatic ring relative to the applied magnetic field. For example, in Smernik et al. (2000a,b), for ^{13}C nuclei in PyOM, we reported SSBs to contain ~10% of signal where the MAS rate was 5kHz and the Larmor frequency was 50 MHz for ^{13}C (i.e. a 200 MHz spectrometer, as NMR spectrometers are described in terms of the Larmor frequency of ^1H , which is ~4x that of ^{13}C). In general, unless rapid MAS (e.g. >10 kHz) is employed, SSBs are observed for ^{13}C NMR of PyOM. There are indeed SSBs in the ^{13}C NMR spectra (totaling ~20% with respect to the isotropic peak) but they were suppressed by TOSS (Total Suppression of Spinning Sidebands, Dixon, 1982), to evaluate the extent of aromatization of the PyOM and the concomitant loss of aliphatic residues. In the original manuscript we had neglected to mention the use of TOSS to remove SSBs – this has now been amended in the Methods and Supplementary Information sections. See Fig. B below for ^{13}C CPMAS spectra with SBBs, which has also been added to the manuscript.

Furthermore, there is a very practical reason for using TOSS in the current study: the PyOM material, being a char formed at 500 °C, is prone to pulse break-through and arcing in the probe. The TOSS sequence, in addition to suppressing spinning side-bands, also suppresses pulse break-through and arcing, both in the ^{13}C and the ^{15}N observe mode. By now showing both spectra and explaining these effects, we take care of questions that readers may have regarding SSBs.

Then, in my experience, 1 ms as contact time is too short for this type of organic systems.

Response: We agree with the referee that choosing the most appropriate contact time is a complicated and critical affair, as it depends on both the rate of cross polarization and the rate of ^1H relaxation in the “rotating frame” ($T_{1\rho}\text{H}$). We have investigated this in detail (Smernik et al., 2002a,b) for ^{13}C NMR of PyOM and the upshot of those investigations was that although it is true that a 1 ms contact time does result in loss of signal for ^{13}C nuclei >4 bonds from their nearest ^1H neighbour, the alternative of increasing the contact time results in even greater loss of signal due to $T_{1\rho}\text{H}$ relaxation, especially for ^{13}C nuclei within ~ 10 nm of an unpaired electron. Thus, a 1 ms contact time was deemed appropriate and the newly collected data shown in Figure C below and added to the Supplementary Information confirms this to be the case.

Also the recycle delays of only 3 s (for CPMAS) and 10 s (for SEDP) are too short. How did the authors chose those parameters?

Response: The influence of $T_{1\rho}\text{H}$ relaxation is easily lost when pure compounds are used in attempting to optimise contact times for “remotely protonated” nuclei. For example, in Smernik and Baldock (2005a), we show that for caffeine, a contact time of 10 ms produces maximum ^{15}N CP signal. However, this only occurs because $T_{1\rho}\text{H}$ is very long in pure crystalline compounds and would not be the case where $T_{1\rho}\text{H}$ is in the range of 3-5 ms, as is the case for organic matter in general (Smernik et al, 2000a,b) and PyOM in particular (Smernik et al. 2002a).

The issue of the recycle delay in CP is quite different – here we judged 3 s to be easily long enough based on previous experience with ^{13}C CP analysis of PyOM (e.g. $T_1\text{H}$ for chars produced at 200-350°C was <100 ms, so a 3-s recycle delay is >30x $T_1\text{H}$, much higher than the recommended 5x $T_1\text{H}$ needed to ensure <1% signal loss to saturation). However, we have now in this revision also directly measured the bulk ^1H relaxation of the PyOM and indeed found it to be less than 100 ms (see Fig. A below). Therefore, a recycle delay of 3 s is more than adequate for CPMAS (see Fig. B below), and in fact, a 1 s recycle delay is sufficient without compromising the signal response. Any shorter length of time may compromise the duty cycle in the probe and damage its workings. Still, we appreciate this concern and have collected several spectra using a range of recycle delay times in order to demonstrate that the recycle delays and contact times used were appropriate. We found that a contact time of 1 ms is indeed optimal because a longer value leads to signal loss (see Fig. C below).

It should be noted that the recycle delay in CP is a function of the longitudinal relaxation of the ^1H population (i.e. $T_1\text{H}$), and so is the same for ^{13}C CP and ^{15}N CP, and is greatly aided by the process of spin diffusion, which facilitates rapid and relatively uniform relaxation of the whole ^1H population. The recycle delay for direct polarization does not have this advantage; it depends on the longitudinal relaxation of the $^{15}\text{N}/^{13}\text{C}$ population (i.e. $T_1\text{N}/T_1\text{C}$) and since both ^{15}N and ^{13}C are isotopically rare, spin diffusion is very inefficient and hence $T_1\text{N}$ and $T_1\text{C}$ are generally much longer and less uniform than $T_1\text{H}$

for complex materials such as PyOM. However, the point of employing the spin echo in SEDP is to facilitate reversion to near-equilibrium magnetisation between scans, and for SEDP a 10 s recycle delay is adequate unlike for “standard” DP, where 100-120 s is more appropriate. We conducted further experiments to test the relaxation behavior of the carbons in PyOM with SEDPMAS (Fig. D), and found that full relaxation of both peaks in the spectrum occurs after ~120s, but that the relative ratio of the two are very similar at 10s and at 120s.

Figure A. ^1H relaxation of oxidized PyOM. This verifies that the T_1 relaxation of the bulk protons in the PyOM sample is less than 100 ms, which shows that a recycle delay of 3 s is more than adequate for ^{13}C CPMAS. The experiment is an array, varying D1 for each time point, the relaxation delay, with $n_s = 4$.

Figure B. ^{13}C CPMAS spectra collected from oxidized PyOM using a recycle delay time of 3 – 20 seconds. This shows that a recycle delay of 3 s is adequate for ^{13}C CPMAS and that increasing the recycle delay time does not change or improve the spectra collected from the PyOM samples, as would be expected when the $T_1\text{H} < 100\text{ms}$.

Figure C. ^{13}C CPMAS-TOSS spectra collected from oxidized PyOM using a contact time of 1 – 5 ms. This verifies that 1 ms is the optimum contact time for observing ^{13}C with CPMAS-TOSS spectra collected from these PyOM samples. The signal decreases when a longer contact time is used.

Figure D. ^{13}C SEDPMAS spectra collected from oxidized PyOM using a relaxation delay of 2 – 150 seconds. Left pane: SEDPMAS relaxation delay experiment. Right pane: Normalized SEDPMAS spectra for peak ratio comparison of spectra collected using a relaxation delay of 10 vs 150 s. These spectra show that full relaxation occurs around ~ 150 s. However, the relative ratio of peaks within each spectrum is similar for spectra collected with a relaxation delay of 10 s and 150 s, suggesting that overall interpretation of the results will not change based on an increase in relaxation delay time.

I think that the only way to get quantitative information is by running direct polarization pulse sequence. However, I understand that this kind of experiment is too time consuming, but the experiments on which the authors base their discussion and conclusions are based on the assumption that all the samples reveal the same relaxing dynamics. This can be or cannot be the case. How can the authors prove this assumption?

Response: As summarized above, we chose the original parameters for NMR quantitation based on substantial previous work with soil organic matter and PyOM. Additionally, the samples used in the

experiments presented here were produced using highly controlled pyrolysis and post-pyrolysis processing (Enders et al., 2012, Bioresource Technology). This resulted in a high degree of sample homogeneity, particularly compared to environmental organic matter samples often studied and reported upon in the literature. The two samples that we compare using ^{13}C NMR spectra in this manuscript have very similar physiochemical properties: both were prepared from the same feedstock under the same conditions and were oxidized to varying degrees, thereby justifying the use of similar NMR data collection parameters. We also conducted new ^{13}C spin echo direct polarization pulse sequence experiments in order to further examine the effect of the recycle delay time on the relaxing dynamics of the particular samples used in our experiments and assess the best parameters for quantitation. It is true that full relaxation in the ^{13}C SEDP experiments occurs around $\sim 120 - 150\text{s}$ (see Fig. D below). However, the relative ratio of the peaks within the same spectrum does not change markedly, so while the signal response is greater with longer recycle delays, this does not change our final interpretation of the spectra. The inclusion of the spin echo (SE) avoids pulse breakthrough especially with such partially conducting materials such as PyOM, and it affords better line shape and baselines in the resultant spectra. We have amended the Methods section and Supplementary Information to reflect these details.

We also conducted new ^{15}N spin echo direct polarization pulse sequence experiments using the same oxidized PyOM sample that was exposed to ^{15}N - NH_3 and used to collect the original NMR spectra presented in our paper. SEDPMAS for ^{15}N is even more necessary, because on our NMR instruments, the pulse breakthrough is more exaggerated at the much lower frequency and compromises signal quality. The original recycle delay of 300 s for the ^{15}N SEDPMAS spectrum was taken from published work (Smernik & Baldock). Fig. E shows the effect of relaxation delay length on ^{15}N SEDPMAS NMR spectra. Fig. F shows the relative integral area of the three major functional groups represented in these spectra. These new data confirm that a relaxation delay of 300 s is sufficient to achieve full relaxation and quantify relative concentrations of N functional groups in oxidized PyOM that has been exposed to NH_3 .

Figure E. ^{15}N SEDPMAS NMR spectra collected from oxidized PyOM after treatment with ^{15}N - NH_3

using a relaxation delay of 5 – 400 s. These spectra show that a 300 s relaxation delay is sufficient to achieve full relaxation.

Figure F. Relative integral area of spectral features consistent with N functional groups represented in ^{15}N SEDPMAS NMR spectra collected from oxidized PyOM after treatment with $[^{15}\text{N}]\text{-NH}_3$ using a relaxation delay of 5 – 400 s. The relative integral area representing each N functional group changes when spectra are collected using a relaxation delay of less than 100 s. However, spectra collected using a relaxation delay of 100 s or greater result in similar relative integral areas for each N functional group. This shows that a relaxation delay time of 300 s can be used for relative quantitation of N functional groups.

In summary, the referee's questions relating to experimental details and quantitation in solid-state NMR analysis are very valid, and we believe our analyses have dealt with these in the best way possible and address the concerns that we anticipate other readers may have.

REVIEWERS' COMMENTS:

Reviewer #5 (Remarks to the Author):

I have evaluated the new version of the manuscript and the authors' answers. I am satisfied by their corrections, thus I think that the paper can be accepted in its present form.

I have only a point which I find very useful for a scientific discussion with the entire scientific community. The authors wrote in their answer:

<This is indeed, on the surface, a surprising outcome that has not been considered by most scientists investigating PyOM-N interactions and may direct us to rethink PyOM material science and N biogeochemistry on local and global scales. It should be noted that the reaction of amines (including ammonia as the simplest case) with carboxylic acids to form amides is very well known—it is, in fact, the basis of protein synthesis from amino acids. It is also well established that in some situations further reactions, including condensation and cyclization and aromatization to form N-aromatics, are also possible. The surprising thing, at face value, is that this might also occur rapidly at the ambient environmental temperature under seemingly benign conditions. However, the Paal-Knorr pyrrole synthesis reaction—which produces pyrroles through the condensation of a dicarbonyl compound with an amine or ammonia—is thought to be responsible for N heterocycle formation between secondary organic aerosols and ammonia or amines in the atmosphere. Furthermore, although we have not investigated the specific chemical reactions ourselves yet, we have searched for mechanisms in the literature. See below for some published papers that include diagrams of proposed reactions between NH₃ and organic compounds>

As far as I understand, the authors are suggesting something that they did not verify directly. In fact, they base their suggestion on some references. This is a clear evidence that the manuscript must be published because this kind of hypothesis must be the object of discussion of the whole scientific community which can either support or deny what the authors wrote.

Reviewer #5 (Remarks to the Author):

I have evaluated the new version of the manuscript and the authors' answers. I am satisfied by their corrections, thus I think that the paper can be accepted in its present form.

I have only a point which I find very useful for a scientific discussion with the entire scientific community. The authors wrote in their answer:

<This is indeed, on the surface, a surprising outcome that has not been considered by most scientists investigating PyOM-N interactions and may direct us to rethink PyOM material science and N biogeochemistry on local and global scales. It should be noted that the reaction of amines (including ammonia as the simplest case) with carboxylic acids to form amides is very well known—it is, in fact, the basis of protein synthesis from amino acids. It is also well established that in some situations further reactions, including condensation and cyclization and aromatization to form N-aromatics, are also possible. The surprising thing, at face value, is that this might also occur rapidly at the ambient environmental temperature under seemingly benign conditions. However, the Paal-Knorr pyrrole synthesis reaction—which produces pyrroles through the condensation of a dicarbonyl compound with an amine or ammonia—is thought to be responsible for N heterocycle formation between secondary organic aerosols and ammonia or amines in the atmosphere. Furthermore, although we have not investigated the specific chemical reactions ourselves yet, we have searched for mechanisms in the literature. See below for some published papers that include diagrams of proposed reactions between NH₃ and organic compounds>

As far as I understand, the authors are suggesting something that they did not verify directly. In fact, they base their suggestion on some references. This is a clear evidence that the manuscript must be published because this kind of hypothesis must be the object of discussion of the whole scientific community which can either support or deny what the authors wrote.

Response: Thank you for your feedback on our manuscript. We agree that this is an interesting point and that a discussion with the scientific community would be very useful. We have added a paragraph to the Discussion section of the manuscript in order to facilitate this.